# G2DC-PL+ A gridded 2 km daily climate dataset for the union of the Polish territory and the Vistula and Odra basins

Mikołaj Piniewski[1], Mateusz Szcześniak[2], Ignacy Kardel[1], Somsubhra Chattopadhyay[1], and Tomasz Berezowski[3]

[1]Department of Hydrology, Meteorology and Water Management, Warsaw University of Life Sciences, Nowoursynowska 166, 02-787 Warsaw, Poland
[2]Independent researcher
[3]Faculty of Electronics, Telecommunication and Informatics, Gdansk University of Technology, Gabriela Narutowicza 11/12, 80-233 Gdansk, Poland

**Correspondence:** M. Piniewski (mikolaj_piniewski@sggw.edu.pl)

**Abstract.** G2DC−PL+ − a gridded 2 km daily climate dataset for the union of the Polish territory and the Vistula and Odra basins is an update and extension of the CHASE-PL Forcing Data - Gridded Daily Precipitation & Temperature Dataset - 5km (CPLFD-GDPT5). The latter was the first publicly available, high-resolution climate forcing dataset in Poland, used for a range of purposes including hydrological modelling and bias correction of climate projections. While the spatial coverage of the new dataset remained the same, it has undergone several major changes: (1) the time coverage was increased from 1951-2013 to 1951-2019; (2) its spatial resolution increased from 5 to 2 km; (3) the number of stations used for interpolation of temperature and precipitation approximately doubled; (4) in addition to precipitation and temperature, the dataset consists of relative humidity and wind speed data. The main purpose for developing this product was the need for long-term areal climate data for earth-system modelling, and particularly hydrological modelling. Geostatistical methods (kriging) were used for interpolation of the studied climate variables. The kriging cross-validation revealed improved performance for precipitation compared to the original dataset expressed by the median of the root mean squared errors standardised by standard deviation of observations (0.59 vs. 0.79). Kriging errors were negatively correlated with station density only for the period 1951-1970. RMSEsd values were equal to 0.52 and 0.4 for minimum and maximum temperature, respectively, suggesting a small to moderate improvement over the original dataset. Relative humidity and wind speed exhibited lower performance, with median RMSEsd equal to 0.82 and 0.87, respectively. The dataset is openly available from the 4TU Centre for Research Data at https://doi.org/10.4121/uuid:a3bed3b8-e22a-4b68-8d75-7b87109c9feb (Piniewski et al., 2020).

## 1 Introduction

Recent decades have witnessed a substantial improvement in atmospheric numerical weather prediction and climate model simulations. Precipitation and air temperature data at high spatial and temporal resolution indeed serve as major input in modelling earth and environment. One of the major applications of such data data is in distributed hydrological modelling at various spatio-temporal scales (Chattopadhyay et al., 2017). Precipitation is a critical component in rainfall–runoff models such

as SWAT (Arnold et al., 1998), WetSpa (Liu and De Smedt, 2004) or TOPMODEL (Beven et al., 1995). Precipitation also plays a major role in flood-drought assessment or provision of ecosystem services (Abbaspour et al., 2015). Temperature influences evaporation, transpiration, and the overall water demand. It is therefore also crucial to provide high resolution temperature data in hydrological models. Methods of estimation of actual evapotranspiration such as Hargreaves or Pennman-Monteith that requires temperature data, are commonly used in several models (Hargreaves, 1975; Wang and Tedesco, 2007).Climate change across the landscape has significant spatiotemporal variations which are often not uniform or consistent (Hayhoe et al., 2008). Spatial heterogeneity in the distribution of earth surface features including physical variables such as land cover, soil moisture as well as landscape properties such as slope, elevation interact with large scale climate which in turn determines microscale climate (Dobrowski et al., 2009). Herein lies the significance of fine scale gridded analysis to study local climate and variability

Spatial resolution of global datasets including gridded precipitation and temperature (Sheffield et al., 2006; Dee et al., 2011; Schamm et al., 2014; Weedon et al., 2014) is variable, with the highest resolution of 0.25x0.25 degrees translating to 28x28 km at the equator and 28x14 km at 60 degrees North. Some of the recent developments in terms of finer resolution at the global scale include Terra Climate (Abatzoglou et al., 2018), providing monthly gridded data at 4 km resolution. Large scale hydrological modelling studies often employ these global datasets (Haddeland et al., 2011; Li et al., 2013; Abbaspour et al., 2015). Previous research has shown that when the study area is smaller than one grid cell of the data, such coarser resolution is not adequate. Therefore, local meteorological gridded datasets are receiving growing attention with country-wide applications recently reported for the UK (Hollis et al., 2018), Iberia (Herrera et al., 2019), Norway (Lussana et al., 2018), Spain (Serrano-Notivolli et al., 2017), China Peng et al. (2019). These products offer a greater advantage for hydrological modelling at the local scale compared to their global counterparts, as shown for a large dataset of French catchments and the conceptual model GR4J (in French, modèle du Génie Rural à 4 paramètres Journalier) (Raimonet et al., 2017).

Gridded datasets are constructed by interpolation from national meteorological networks. It is still unclear, however, how to choose the optimal method for spatial interpolation of these meteorological variables. The geostatistical (kriging) and inverse distance weighted (IDW) methods are generally quite popular. Some of the recent studies have used advanced interpolation techniques such as regression kriging (Brinckmann et al., 2016), iterative optimal interpolation (Lussana et al., 2018) and the area-averaged three-dimensional (AA-3D) interpolation method (Herrera et al., 2019). Kriging outperformed IDW and Thiessen polygons as evaluated by Szcześniak and Piniewski (2015) in a hydrological modelling study for several medium-sized catchments in Poland. Kriging is indeed still being used as the interpolation method for precipitation and air temperature with satisfactory results quantified by correlation coefficients or root mean squared errors (Carrera-Hernández and Gaskin, 2007; Hofstra et al., 2008; Ly et al., 2011; Herrera et al., 2012; Brinckmann et al., 2016; Herrera et al., 2019). A broad range of kriging types exists, including: ordinary kriging (assumes constant mean), universal kriging (removal of trend based on spatial coordinates), kriging with external drift (mean is dependent on external variable, e.g. elevation map), co-kriging (estimates a variable based on its values and values of other variables) and others. Selection of the most appropriate kriging method is variable dependent (by studying phenomena responsible for observations of the variable, e.g. relations with elevation) and case-study dependent (by investigating whether any trend is observed at a given spatial and temporal scale, e.g. seasonal and geographical relations with climate).

Berezowski et al. (2016) described the CHASE−PL Forcing Data − Gridded Daily Precipitation & Temperature Dataset − 5km (CPLFD−GDPT5) product, which provided input data for environmental modelling in the area defined as the union of the Polish territory and the Vistula and Odra basins (hereafter denoted as 'PL+'). It was the first publicly available, high-resolution climate forcing dataset in Poland. It has been widely used for a range of purposes, including: (1) bias correction of EURO-CORDEX projections (Mezghani et al., 2017); (2) forcing data for hydrological models (Piniewski et al., 2017), (Terskii et al., 2019), a water quality model (Marcinkowski et al., 2017) and a simple phenology model (Marcinkowski and Piniewski, 2018), (3) a streamflow trend detection study (Somorowska, 2017), (4) a hydro-ecological classification (Piniewski, 2017), and (5) ecological modelling (O'Keeffe et al., 2018). Furthermore, the same methodological workflow as in Berezowski et al. (2016) was used to prepare hydrological model forcing in Berezowski et al. (2019). Besides a natural need to carry out a periodic update of the dataset, a number of other potential product improvements have been identified over recent years, such as increasing spatial resolution, adding new variables and increasing the number of stations used for interpolation . In this paper, we describe a new, updated product called G2DC−PL+ (a gridded 2 km daily climate dataset for the PL+ area), specifically pointing to differences between itself and its predecessor, CPLFD−GDPT5.

## 2 Key features of the update

While the spatial domain of the G2DC-PL+ dataset remained the same as in the original CPLFD−GDPT5 dataset, i.e. the PL+ area (Fig. 1), the dataset has undergone several major changes: (1) the temporal range has been extended from 1951-2013 to 1951-2019; (2) the spatial resolution has increased from 5 km to 2 km; (3) the number of stations used for interpolation of temperature and precipitation approximately doubled; (4) in addition to precipitation and temperature, the new dataset consists of relative humidity and wind speed.

### 2.1 Temporal range

The analysis cover the period 2014-2019, the six year extension (2014-2019) makes the dataset applicable for studying recent earth-systems phenomena. Poland as well as the neighbouring countries in Central and Eastern Europe have encountered three major droughts in 2015, 2018 and 2019. Inclusion of these years in the dataset will be useful for in-depth drought assessments and will help to constrain hydrological models (Pfannerstill et al., 2014).

### 2.2 Interpolated variables

While the previous dataset included only two climate variables,namely temperature and precipitation, the updated version includes two new ones; relative humidity and wind speed (average daily values). We originally planned to include solar radiation as well, but due to a low number of stations available in Poland (below 30) and much shorter temporal availability of data, we concluded there would be little benefit in doing so. A viable alternative for solar radiation data is offered in the E−OBS gridded dataset distributed by ECA&D. One of the major benefits of using relative humidity, wind speed and solar radiation data in addition to air temperature data is that it allows for using the energy-based Penman-Monteith method for potential evap-

otranspiration (PET) calculation instead of a temperature based method, such as Hargreaves. PET is a key input in hydrological models, and its role is of particular importance in the ever-growing climate change impact studies. Some authors advocate the use of a fully physically-based formulation of PET rather than temperature-based PET methods. The uncertainty of using more simple methods in climate change impact studies is huge and similar in its magnitude to the uncertainty of General Circulation Models (Hosseinzadehtalaei et al., 2016) or emission scenarios (Williamson et al., 2016).

### 2.3 Number of stations

At the time of the development of the CPLFD-GDPT5 product (2015-2016), the data sharing policy of the key institution owning climate data in Poland, the Institute of Meteorology and Water Management (IMGW−PIB) required concluding individual agreements between the data holder and the user. Due to this , only a subset of all available historical station data could be obtained at that time. Since 2018, IMGW−PIB has changed its policy and made the climate data freely available for research purposes at the https://dane.imgw.pl/ website. It has allowed us to download and use all available data for the entire period of interest, namely 1951-2019. The number of IMGW−PIB precipitation stations with available data (approaching 1500) was the highest in the early 1990's. The number is more than double the maximum number of respective stations in the former dataset. The number of IMGW−PIB stations at the beginning and end of the analysed period, however was significantly lower (600-700) (Figure 2). With regard to other variables, the temporal trend of data availability was the same as for precipitation, but the total numbers were significantly lower, reaching approximately 250 in early 1990s (cf. Figures 3-5). Spatial distribution of used stations is shown in Figures 6-9.

The major source of data from outside Poland was the European Climate Assessment & Dataset (ECA&D). Also in this case, a large, almost 10−fold increase in data availability was observed for the area of interest. Among countries neighbouring with Poland, station density was the highest in Germany. The third, least abundant data source was the National Oceanic and Atmosphere Administration - National Climatic Data Center (NOAA-NCDC), but in this case the number of stations did not change much.

### 2.4 Spatial resolution

Considering increasing computing power and storage capacity, an increase in dataset spatial resolution is a natural choice. The original 5 km resolution was not sufficiently high, in particular in mountainous areas in the south of study area. The output resolution of 2 km is of particular importance for precipitation because it is characterised by higher spatial variability. A number of gridded precipitation datasets at comparably high resolution, developed predominantly for hydrological applications, were issued recently (Duan et al., 2016; Laiti et al., 2018; Lewis et al., 2018). Another reason was that temperature, humidity and wind speed were interpolated using kriging with external drift method, in which elevation was used as a co-variable. Elevation at 2 km is much more accurate than at 5 km resolution, especially in high altitude areas, so this should be a clear, although indirect, benefit of this approach.

Even though station density for variables other than precipitation was not very high (see Fig. 7-9), for practical reasons we have decided to set a uniform resolution of 2 km throughout the entire product.

## 2.5 Unchanged properties of the updated dataset

Other methodological features that remained unchanged between the current and previous versions are as follows:

1. The projected coordinate system for all gridded data was PUWG−92.

2. All organisations from which we have compiled the data conduct quality control check for raw data before making them publicly available.

3. The time frequency for all variables was daily.

4. Correction for precipitation undercatch was carried out by means of the Richter method (Richter, 1995) recognized by the World Meteorological Organization. A map showing values of coefficient $b$ representing the effect of wind exposition of the measurements site is presented in Figure 10). We followed the same, simplified criteria for dividing stations into those with 'low' and 'medium' shielding as Berezowski et al. (2016). Stations located above 400 m a.s.l. and those lying within a 40 km buffer from the coast were assigned to a 'medium' shielding category.

   The values of b were set as for "medium shielding" for all stations apart from those in the mountains or close to the coast, where b were set as for "low shielding" (Fig. 5). The rationale behind assigning different values of b for different location lies in the fact that wind speed is generally higher in mountains and at the seaside than in the lowlands.

5. The applied procedure for filling in '0' values to precipitation time series for a subset of IMGW−PIB stations was similar to that in (Berezowski et al., 2016). Due to improved meta-data reporting by IMGW−PIB, however, it has been applied much less frequently than in the original dataset. Removal of suspicious values from the time series was also carried out in a way similar as before.

6. Minimum and maximum temperatures were interpolated with kriging with external drift and precipitation with a combination of universal and indicator kriging. The exponential variogram model was used in each case with the variogram parameters estimated automatically for each daily kriging with the weighted least squares fit (Pebesma, 2004). The block kriging approach was applied with block size equal to the output square grid size, i.e., 2 km. The two new variables, namely relative humidity and wind speed, were interpolated using the same method as for temperature.

7. A "leave-one-out" cross-validation was performed daily for all stations, i.e., each station was removed from the sample one at a time and the remaining stations were used to predict the value of the missing station. There is one small deviation from the previous version of cross-validation affecting only precipitation variable, based on the study of Berndt and Haberlandt (2018). Because a 6.25-fold increase in spatial resolution and more than 2-fold increase in the number of stations caused a significant increase in cross-validation calculation time, we have decided to apply cross-validation only for days with precipitation above a 1 mm threshold which allowed to speed up the calculation process in a satisfactory way.

8. The interpolation errors were quantified using : (1) Pearson's correlation coefficient ($\rho$) and (2) root mean squared error normalized to standard deviation of the observed data:

$$\text{RMSEsd} = \frac{\sqrt{\frac{1}{N}\sum_{i=1}^{N}\left(Y_i - \hat{Y}_i\right)^2}}{\sigma_Y}$$

where $Y$ and $\hat{Y}$ are respectively the observed and interpolated values of a given variable, $N$ is the number of observations (number of stations in the spatial approach or number of days in the temporal approach) and $\sigma_Y$ is the standard deviation of observations.

The cross-validation was conducted on both temporal and spatial scales. On the temporal scale the errors were calculated for each day from all stations having data on this day. For all variables standard deviation used in calculation of RMSEsd performance metrics was calculated for each Julian day separately. We will hereafter refer to the temporal scale indices as $\rho_t$ and $\text{RMSEsd}_t$. On the spatial scale the errors were calculated for each station from all of a station's available daily values. We will hereafter refer to the spatial scale indices as $\rho_s$ and $\text{RMSEsd}_s$.

The reader is referred to the study of Berezowski et al. (2016) for additional information related to the above mentioned aspects.

## 3 Cross-validation

### 3.1 Precipitation

According to daily $\rho_t$ statistics for precipitation 75% of $\rho_t$ values are higher than 0.53 (0.47[1]) and $\text{RMSEsd}_t$ values are lower than 0.69 (0.93), respectively (Table 1). Median $\rho_t$ is 0.66 (0.65) and median $\text{RMSEsd}_t$ is 0.59 (0.79). The fraction of RMSEsd values larger than one dropped from 14.2 % to 0.16 %.

We also quantified the effect of modifying the method of calculation of $\text{RMSEsd}_t$ (see subsection 2.5). For a subset of approximately 10 % of years $\text{RMSEsd}_t$ values were calculated using both the new, faster approach, involving removal of dates with low precipitation, and the original approach. We concluded that removal of low precipitation data led to a slight improvement (average difference between $\text{RMSEsd}_t$ values equal to 0.04). The scale of this improvement, however is much lower than the overall improvement discussed in the previous paragraph.

A strong negative correlation exists between the median of daily $\text{RMSEsd}_t$ values and the number or available precipitation stations for the first approximately 20 years of the dataset (Figure 11). While the number of stations steadily increased over the period 1971-1990, the cross-validation error in the same period oscillated without any clear trend. Interestingly, the decline in the number of stations after 1991 was associated with the decrease in $\text{RMSEsd}_t$ value.

---

[1]Respective values referring to the cross-validation errors reported in Berezowski et al. (2016) will be herein shown in parentheses.

Like in the original dataset, when considering the $\mathrm{RMSEsd}_s$ (calculated spatially) for all stations, the results show a clear pattern of higher errors at the edge of the interpolation area, particularly in all neighbouring countries except for Germany (Figure 11). Note that Germany features much higher density of stations than other countries neighbouring with Poland, which explains the spatial pattern.

The interpolation error for precipitation expressed in absolute, not-standardized values, i.e. RMSE, equals 1.6 mm (see Table 1 for other statistics).

## 3.2 Temperature

The statistics of daily $\mathrm{RMSEsd}_t$ show the median equal to 0.52 (0.54) for minimum temperature and 0.4 (0.47) for maximum temperature (Table 1). None of the $\mathrm{RMSEsd}_t$ values in any of the cases exceeds the value of 1 (meaning that the root mean squared errors are always below the standard deviation of observations), and all the $\rho_t$ values are positive. The median of $\rho_t$ is 0.72 (0.84) for minimum temperature and 0.84 (0.88) for maximum temperature. This suggests that, the value of $\rho_t$ measuring the collinearity between simulations and observations used to be higher in the former version of the dataset, particularly for minimum temperature.

Inter-annual variability of $\mathrm{RMSEsd}_t$ exhibits quite different behaviour for minimum (Figure 13) and maximum (Figure 14) temperature. In the former case, a sharp decreasing trend, possibly connected to the rising number of stations, can be observed until 1980, followed by an increase until the early 2000's and another decrease afterwards. In the latter case there is a clear negative trend for the entire period, although in its middle (1970-2000) the values were fluctuating around the mean without any trend. This behaviour is not fully consistent with the temporal evolution of $\mathrm{RMSEsd}_t$ for these two variables in (Berezowski et al., 2016), but the previously stated conclusion that kriging errors for temperature are not dependent on the density of the observation network seems to hold true.

In the analysis of $\rho_s$ and $\mathrm{RMSEsd}_s$ for all stations both the minimum and maximum temperature show a rather uniformly distributed values with very few outliers, usually located in the proximity of the edge of the interpolation area (Figures 15-16). It is noteworthy that the source of data (IMGW$-$PIB or international databases) does not influence the errors for temperature as much as for precipitation.

The interpolation error expressed in absolute, not-standardized values, i.e. RMSE, equals 1.33 and 1.11 ° C for minimum and maximum temperature, respectively (see Table 1 for other statistics).

## 3.3 Relative humidity

The median values of $\rho_t$ and $\mathrm{RMSEsd}_t$ for relative humidity were equal to 0.36 and 0.82, respectively (Table 1). Beacuse relative humidity was not included in the first version of the dataset, no prior statistics are available for comparison, as occurred for precipitation and temperature. $\rho_t$ values for relative humidity are generally considerably lower than for temperature and precipitation, pointing to low collinearity between simulations and observations. The same holds true for $\mathrm{RMSEsd}_t$, although the fraction of data with root mean squared errors exceeding one standard deviation is relatively low, reaching 4.4 %.

$\text{RMSEsd}_t$ values fluctuate in a range of 0.75-0.9 for most of the analysed years. Three sub-periods can be distinguished: 1951-1990 with a low, decreasing trend, 1991-2005 with an increasing trend, and 2005-2019 with a decreasing trend (Figure 17). Trends in the first two sub-periods can be related to changes in relative humidity station density (increasing until 1990, then decreasing).

Spatial variability in $\rho_s$ and $\text{RMSEsd}_s$ is much higher for relative humidity than for temperature or precipitation (Figure 18).
The dataset uncertainty evidently increases with elevation, with stations located in the mountainous, southern belt of the study domain showing the lowest $\rho_s$ and the highest $\text{RMSEsd}_s$ values. The proximity of the coast is another possible cause of higher errors. For the great majority of the Polish Plain, which covers the interior part of the study domain, Pearson's correlation exceeds 0.6 and $\text{RMSEsd}_s$ is below 0.7.

The interpolation error for relative humidity expressed in absolute, not-standardized values, i.e. RMSE, equals 0.06 (see
Table 1 for other statistics).

### 3.4  Wind speed

According to Table 1 median values of $\rho_t$ and $\text{RMSEsd}_t$ for wind speed were found to be 0.24 and 0.87. As was the case with relative humidity, wind speed, was not included in the first version of the dataset, hence comparison which was possible for precipitation and temperature was not feasible in this case. $\rho_t$ values for wind speed were found to be substantially lower than
for other studied variables. Similar pattern was noticed for $\text{RMSEsd}_t$, although the fraction of data with root mean squared errors exceeding one standard deviation is very low, reaching 1.2 %.

Wind speed exhibits lower inter-annual variability of cross-validation errors than other variables (Figure 19). The range of variability during the period 1951-2019 is 0.84-0.92. Furthermore, no trend in the data and no correlation of $\text{RMSEsd}_t$ with station density exists.
Like in the case of relative humidity, spatial variability of $\rho_s$ and $\text{RMSEsd}_s$ was much higher than for temperature or precipitation (Figure 20). The majority of stations with low correlation (below 0.2) is located in Czechia and the great majority of stations with correlation below 0.4 is located at the southern edge of the interpolation domain. $\rho$ values for German stations are higher than for Polish ones. In the case of $\text{RMSEsd}_s$ spatial variability is much lower, but several outliers are also located in the south.
The interpolation error for wind speed expressed in absolute, not-standardized values, i.e. RMSE, equals 1.56 m/s (see Table 1 for other statistics).

### 4  Consistency with climatic data

Berezowski et al. (2016) compared the consistency of the CPLFD−GDPT5 dataset with maps of climatic statistics for the period 1971-2000 provided by IMGW−PIB. The comparison revealed a high level of consistency as well as certain differences
that could be attributed to different data processing methods. Here, we have updated the previous set of these 'comparison maps', and found only a minor difference in climatic means. Therefore, the previous conclusions remain valid. One difference

is that due to a larger number of stations in the northern part of Poland close to the Baltic Sea coast, the updated dataset predicts higher long-term mean precipitation for this area than the previous one. Figures 21, 22 and 23 demonstrate the spatial pattern of temperature, precipitation, relative humidity and wind speed respectively during 1990-2019 from the G2DC-PL+ dataset.

In addition, we present a supplementary analysis focused on the comparison of maps of climatic statistics for selected years. One warm and dry year (2015) and one cool and wet year (2017) were selected for comparison. The maps can be found in the Supplement (Figures S1-S6). Although the G2DC–PL+ dataset precipitation is higher than the IMGW−PIB precipitation due to the applied correction for precipitation undercatch, spatial patterns in both dry and wet year remain very similar between both data sources (Figures S1 and S4). The spatial agreement was also very high for minimum temperature (Figures S2 and

S5) as well as maximum temperature (S3 and S6).

## 5    Data availability

The G2DC-PL+ product is available in NetCDF and GeoTIFF formats. The gridded structure of the data and the NetCDF and GeoTIFF data format ensure easy processing in GIS and data analysis software (e.g. R for both NetCDF and GeoTIFF; list of NetCDF manipulation software: http://www.unidata.ucar.edu/software/netcdf/software.html). Example R scripts allowing

reading the data and conducting basic processing can be found in Berezowski et al. (2016)

The data are publicly available in the 4TU Centre for Research Data repository under the DOI: 10.4121/uuid:a3bed3b8-e22a-4b68-8d75-7b87109c9feb (Piniewski et al., 2020). Files with daily data are organized in decades, e.g. 1951-1960, whereas monthly and annual data are stored in single files. The NetCDF files naming convention is *Variable_TimeStep_FromYear_ToYear*.nc (e.g. pre_d_1951_1960.nc). *TimeStep* can be "d" for day, "m" for month, "a" for annual aggregation period. Every NetCDF

file follows the CF-1.0 convention. *Variable* can be: Tmin/Tmax for minimum/maximum air temperature [°C] or Pre for precipitation [kg m$^{-2}$] or Hmd for relative humidity [%] or Wnd for wind speed [m s$^{-1}$].

Each daily grid for each variable is also stored as a separate GeoTIFF file. The naming convention for zipped collections of GeoTIFF files is: *Variable_TimeStep_FromYear_ToYear_tif*.zip "pre" for precipitation, "tmin" for minimum temperature and "tmax" for maximum temperature; "hmd" for relative humidity and "wnd" for wind speed; whereas time is coded as

YYYYMMDD, YYYYMM or YYYY for daily, monthly and annual time step, respectively.

## 6    Conclusions

In the conclusions of the paper byBerezowski et al. (2016) we stated that the dataset update was planned on a three-year basis. It took slightly more time, but the dataset has been updated with very recent data reaching 2019. The product's spatial resolution has also been increased from 5 to 2 km which may appear crucial for some high-resolution applications. Inclusion of two

additional variables, namely relative humidity and wind speed, although associated with higher interpolation errors as shown in this study, is an important step towards using the energy-based Penman-Monteith method for PET estimation in hydrological or agricultural modelling. Taking advantage of a more open data sharing policy of the key climate data provider, IMGW−PIB,

we have also substantially increased the number of stations available for interpolation of temperature and precipitation, which in the latter case has led to a noticeable reduction of interpolation error. This holds promise that future applications of this dataset for hydrological modelling will benefit from better input data, and will therfore deliver more reliable predictions.

*Author contributions.* MP designed the study. TB and MS developed the model code. MS performed the calculations. IK was responsible for dataset preparation for the repository. MP prepared the manuscript with contributions from all co-authors. All co-authors contributed to the editing of the manuscript and to the discussion and interpretation of results.

*Competing interests.* The authors declare that they have no conflict of interest.

*Acknowledgements.* This study was supported financially by the Warsaw University of Life Sciences (SGGW) "Financial Support Mechanism for Researchers and Research Teams" granted in 2019. MP and SC received support from the National Science Centre (NCN) project called RIFFLES "The effect of RIver Flow variability and extremes on biota of temperate Floodplain rivers under multiple pressurES" $(2018/31/D/ST10/03817)$. We also acknowledge the Institute of Meteorology and Water Management – National Research Institute (IMGW-PIB), European Climate Assessment and Dataset (ECA&D) and National Oceanic and Atmosphere Administration – National Climatic Data Center (NOAA-NCDC) for providing meteorological data.

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

**Table 1.** Descriptive statistics for the kriging cross-validation results for precipitation, minimum temperature, maximum temperature, relative humidity and wind speed. "RMSEsd > 1" denotes percentage of RMSEsd values higher than one.

| | Precipitation | | | Minimum temperature | | | Maximum temperature | | | Relative humidity | | | Wind speed | | |
|---|---|---|---|---|---|---|---|---|---|---|---|---|---|---|---|
| | $\rho$ | RMSEsd | RMSE | $\rho$ | RMSEsd | RMSE | $\rho$ | RMSEsd | RMSE | $\rho$ | RMSEsd | RMSE | $\rho$ | RMSEsd | RMSE |
| Minimum | 0 | 0.11 | 0 | 0.02 | 0.16 | 0.53 | 0.05 | 0.15 | 0.6 | 0 | 0.24 | 0.02 | 0 | 0.48 | 0.66 |
| 1st quartile | 0.53 | 0.48 | 0.92 | 0.62 | 0.44 | 1.11 | 0.77 | 0.35 | 0.99 | 0.18 | 0.71 | 0.05 | 0.17 | 0.82 | 1.24 |
| Median | 0.66 | 0.59 | 1.64 | 0.72 | 0.52 | 1.33 | 0.84 | 0.40 | 1.11 | 0.36 | 0.82 | 0.06 | 0.24 | 0.87 | 1.56 |
| 3rd quartile | 0.77 | 0.69 | 2.48 | 0.8 | 0.62 | 1.6 | 0.88 | 0.47 | 1.25 | 0.52 | 0.92 | 0.07 | 0.33 | 0.91 | 1.98 |
| Maximum | 0.99 | 1.24 | 11.9 | 0.98 | 1 | 3.62 | 0.98 | 0.99 | 2.97 | 0.94 | 1.29 | 0.16 | 0.77 | 1.1 | 4.35 |
| RMSEsd > 1 | – | 0.16 % | – | – | 0 % | – | – | 0 % | – | – | 4.4 % | – | – | 1.2 | – % |

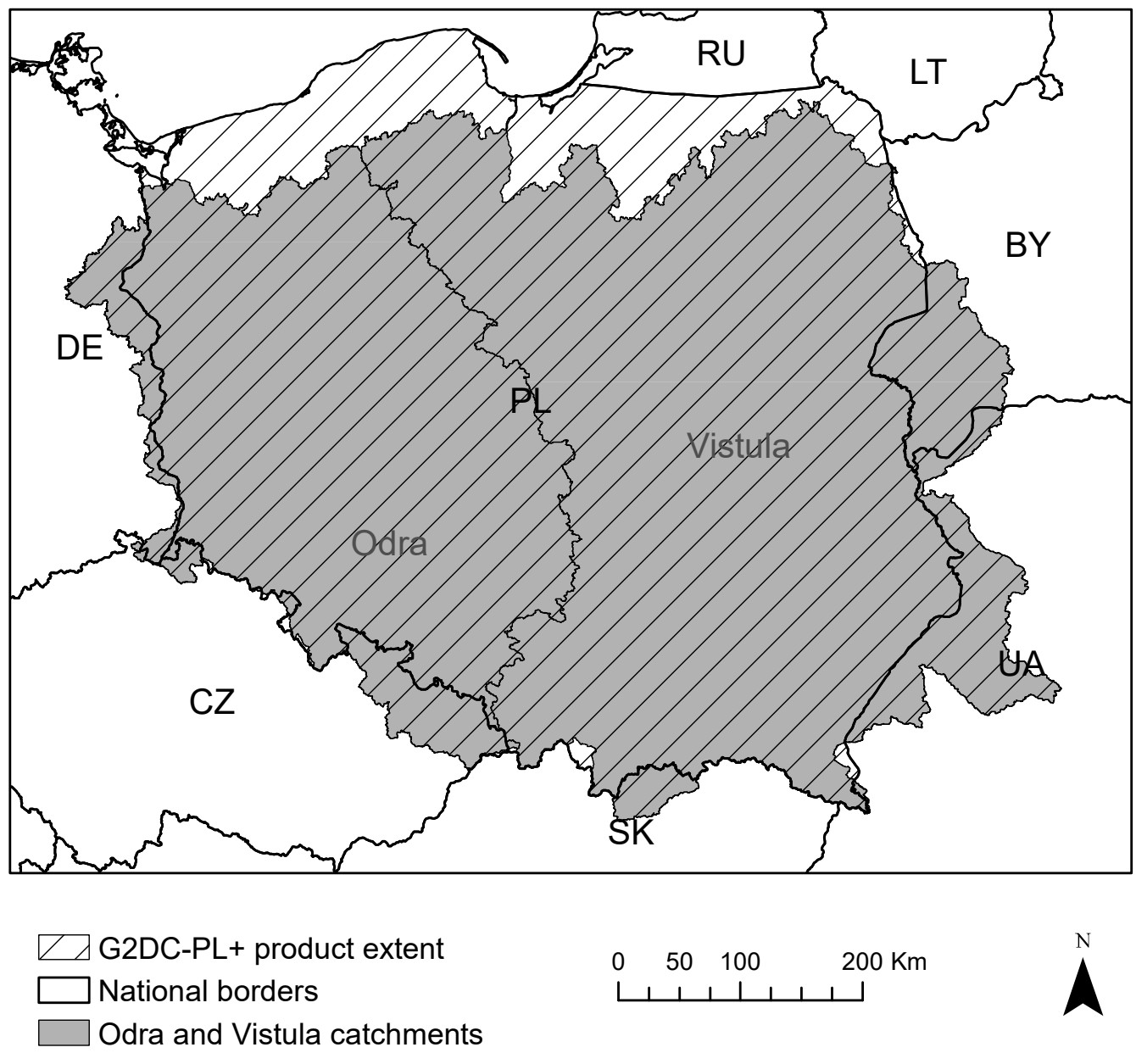

**Figure 1.** The spatial extent for the G2DC-PL+ dataset. Countries are labelled with black national codes. The Odra and Vistula basins are labelled in grey.

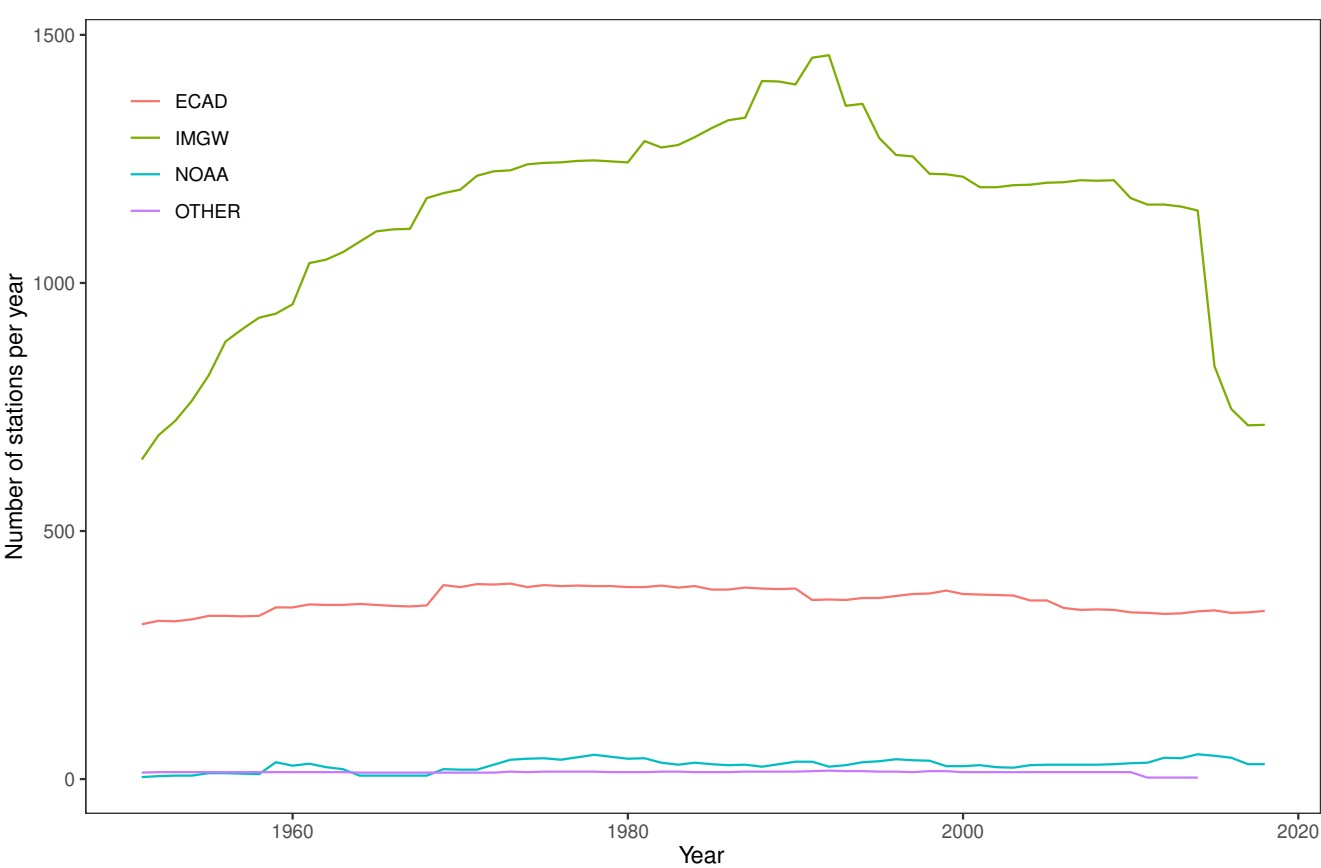

**Figure 2.** Number of meteorological stations for precipitation observations per year from 1951 to 2019.

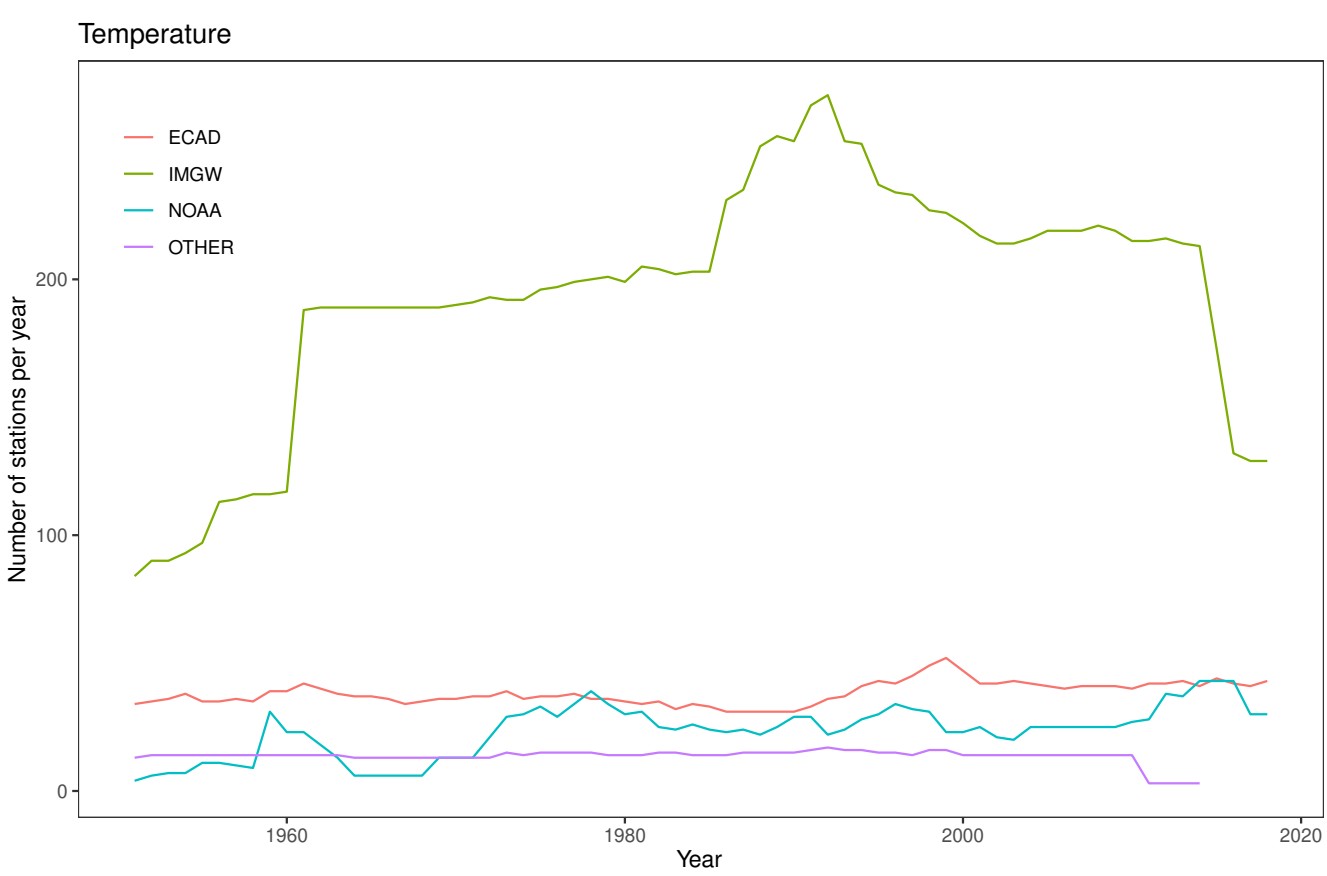

**Figure 3.** Number of meteorological stations for temperature observations per year from 1951 to 2019.

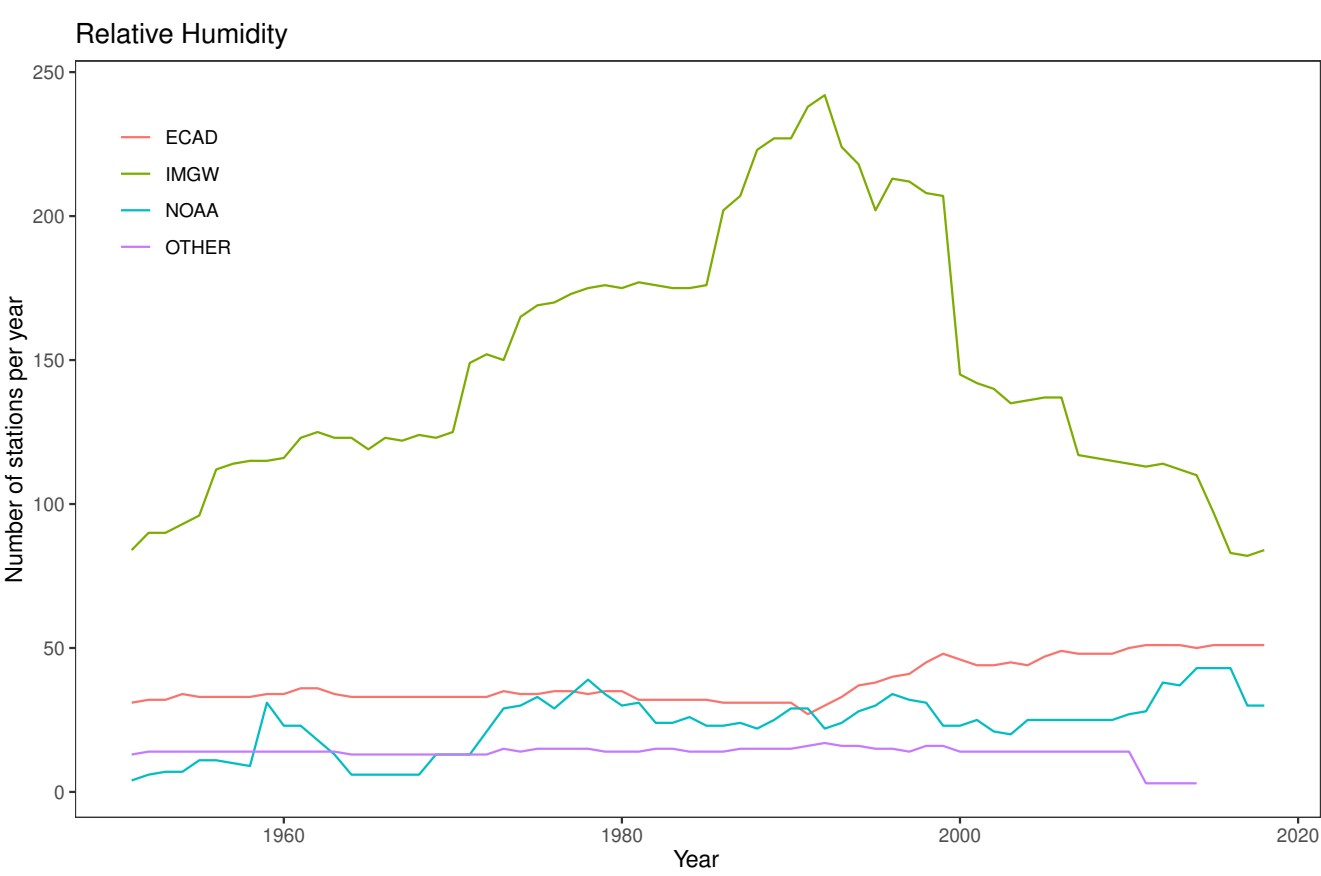

**Figure 4.** Number of meteorological stations for relative humidity observations per year from 1951 to 2019.

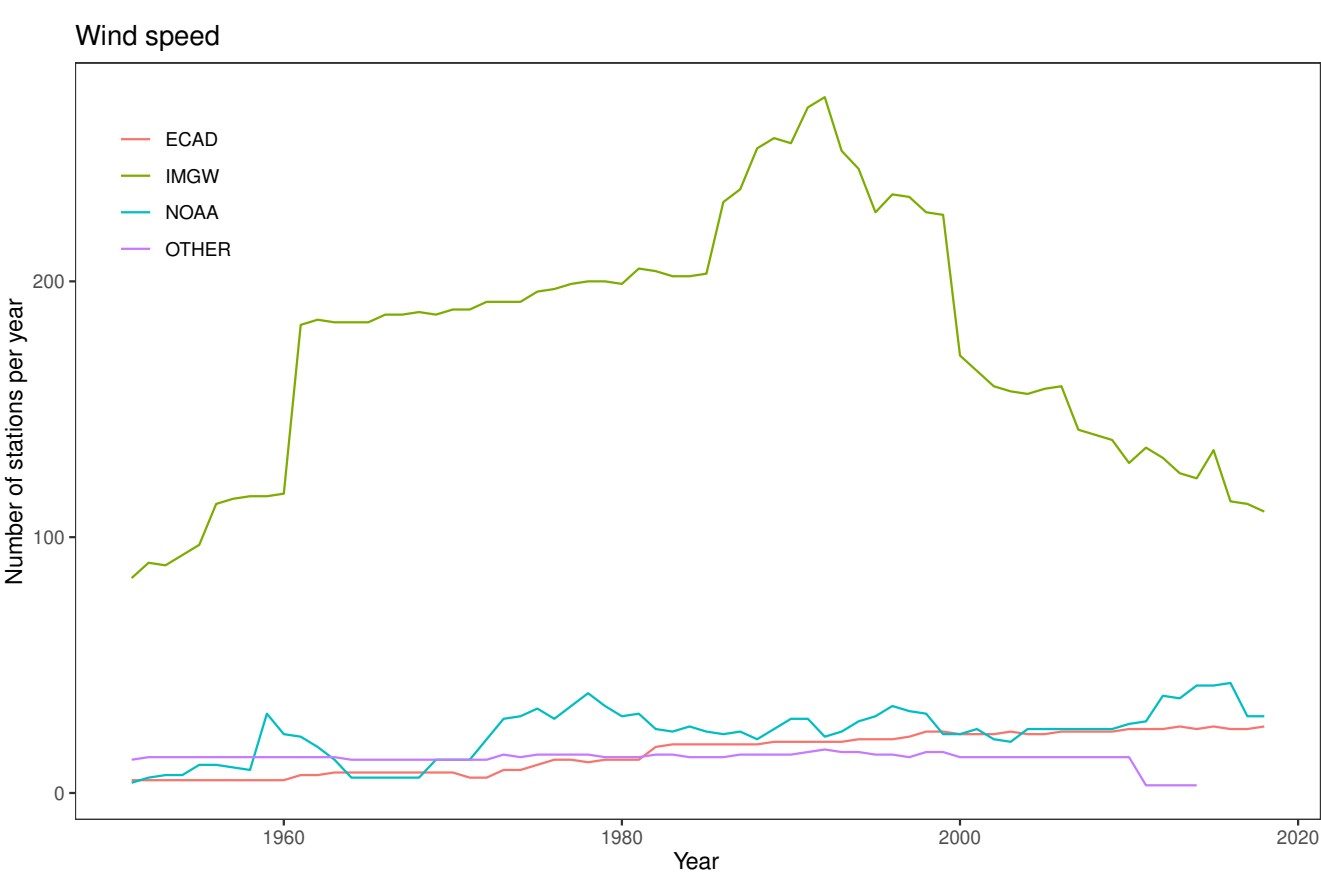

**Figure 5.** Number of meteorological stations for wind speed observations per year from 1951 to 2019.

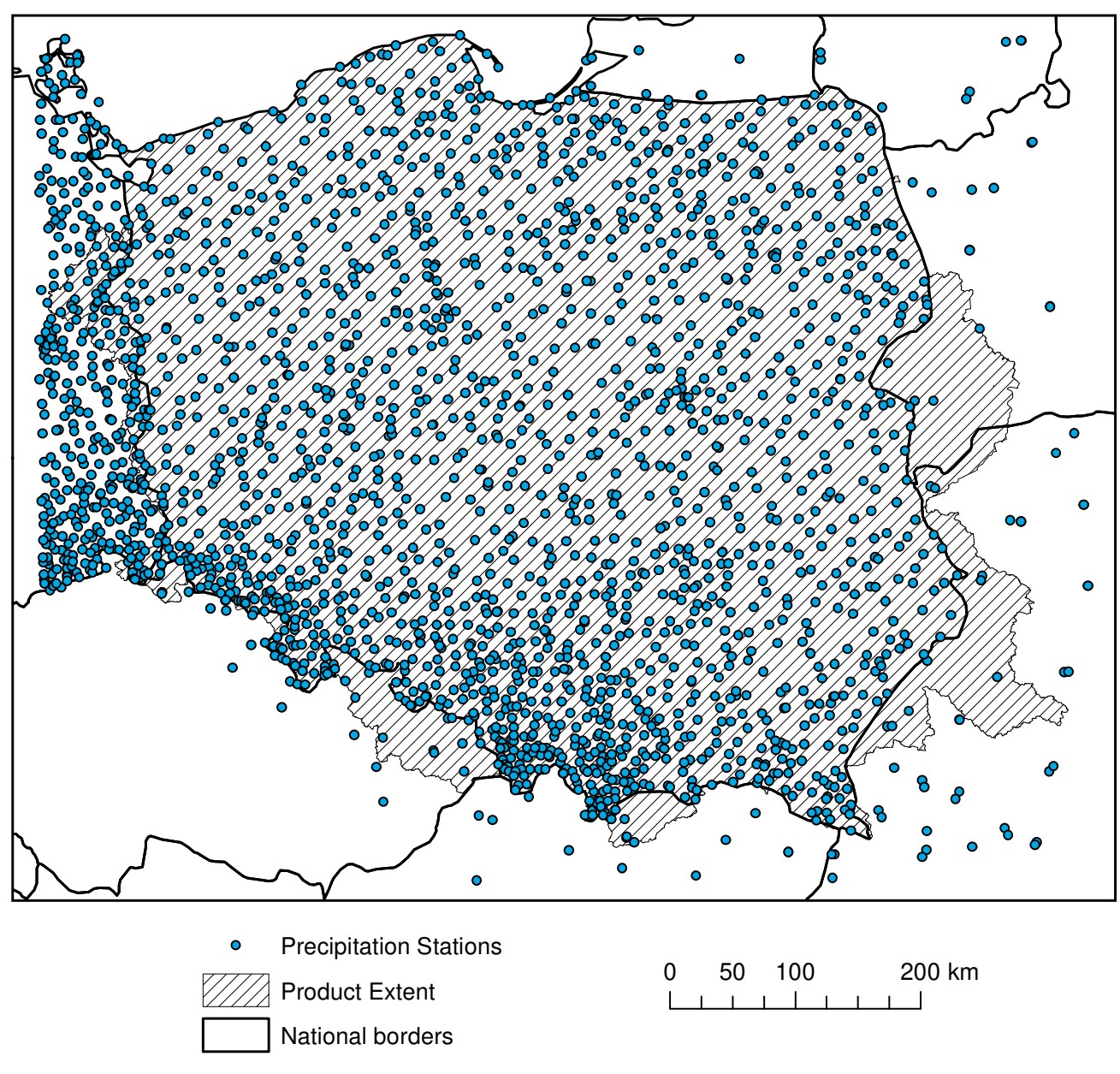

**Figure 6.** Spatial distribution of precipitation stations used for interpolation of the G2DC-PL+ product.

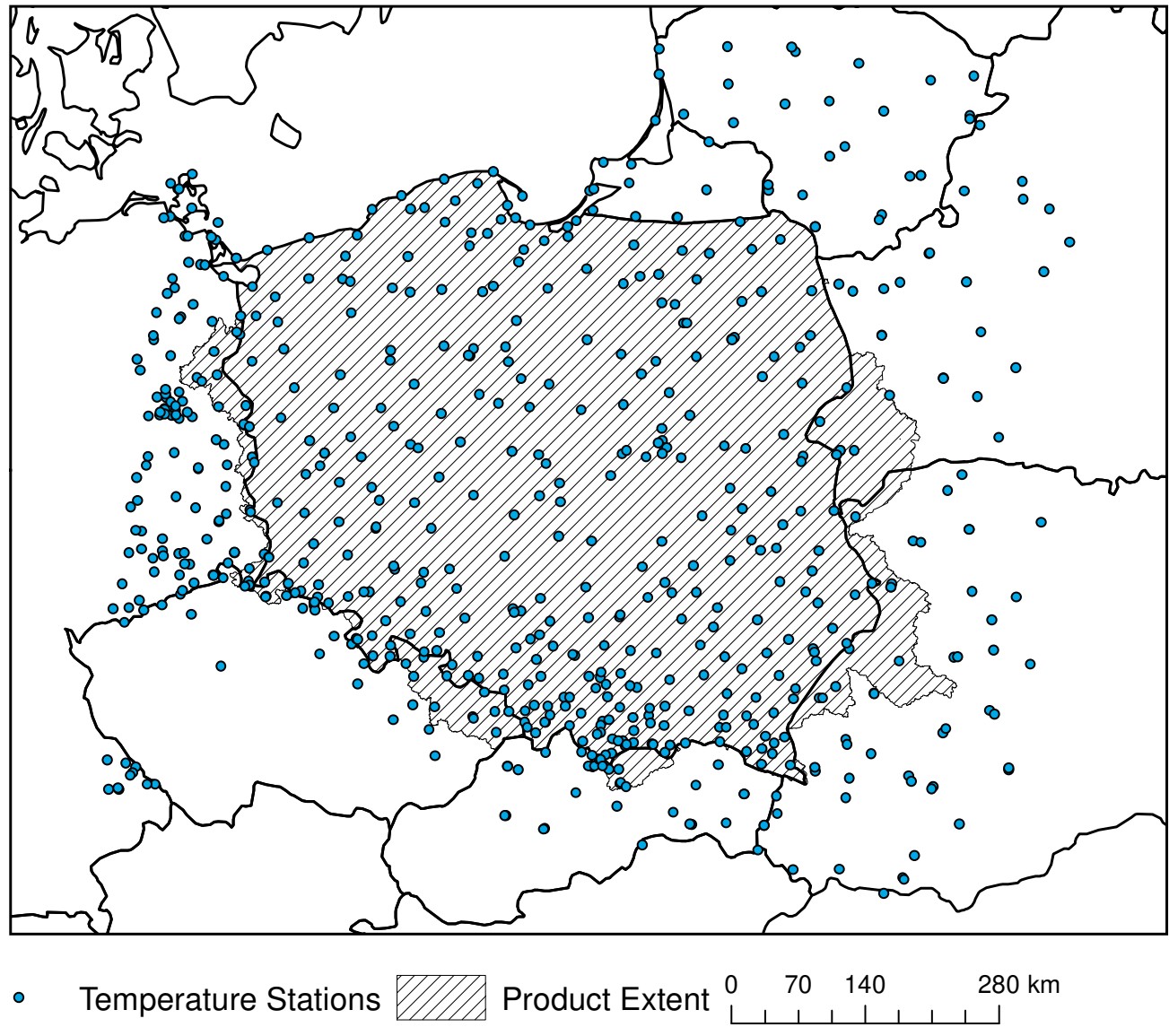

**Figure 7.** Spatial distribution of temperature stations used for interpolation of the G2DC-PL+ product.

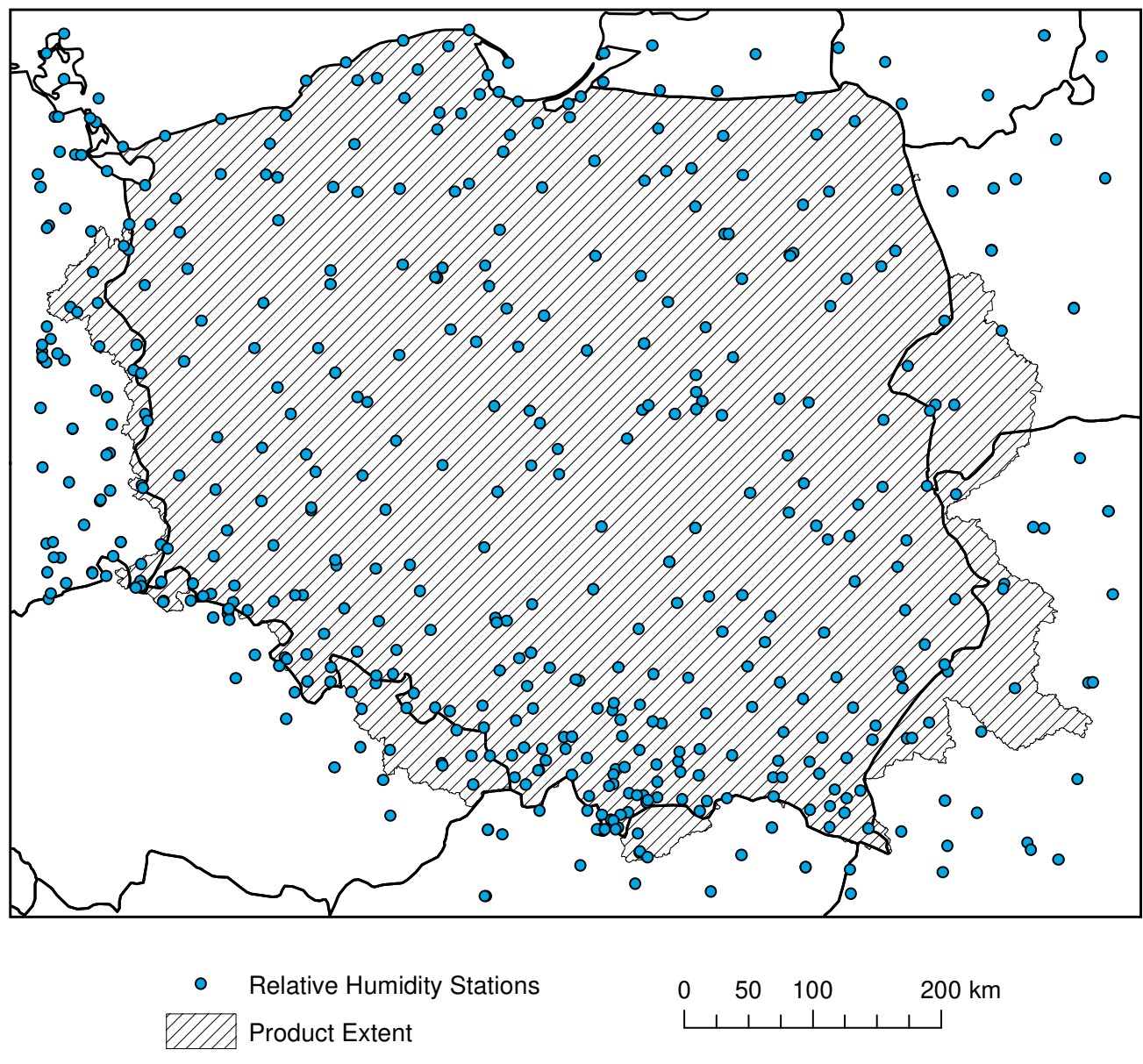

**Figure 8.** Spatial distribution of relative humidity stations used for interpolation of the G2DC-PL+ product.

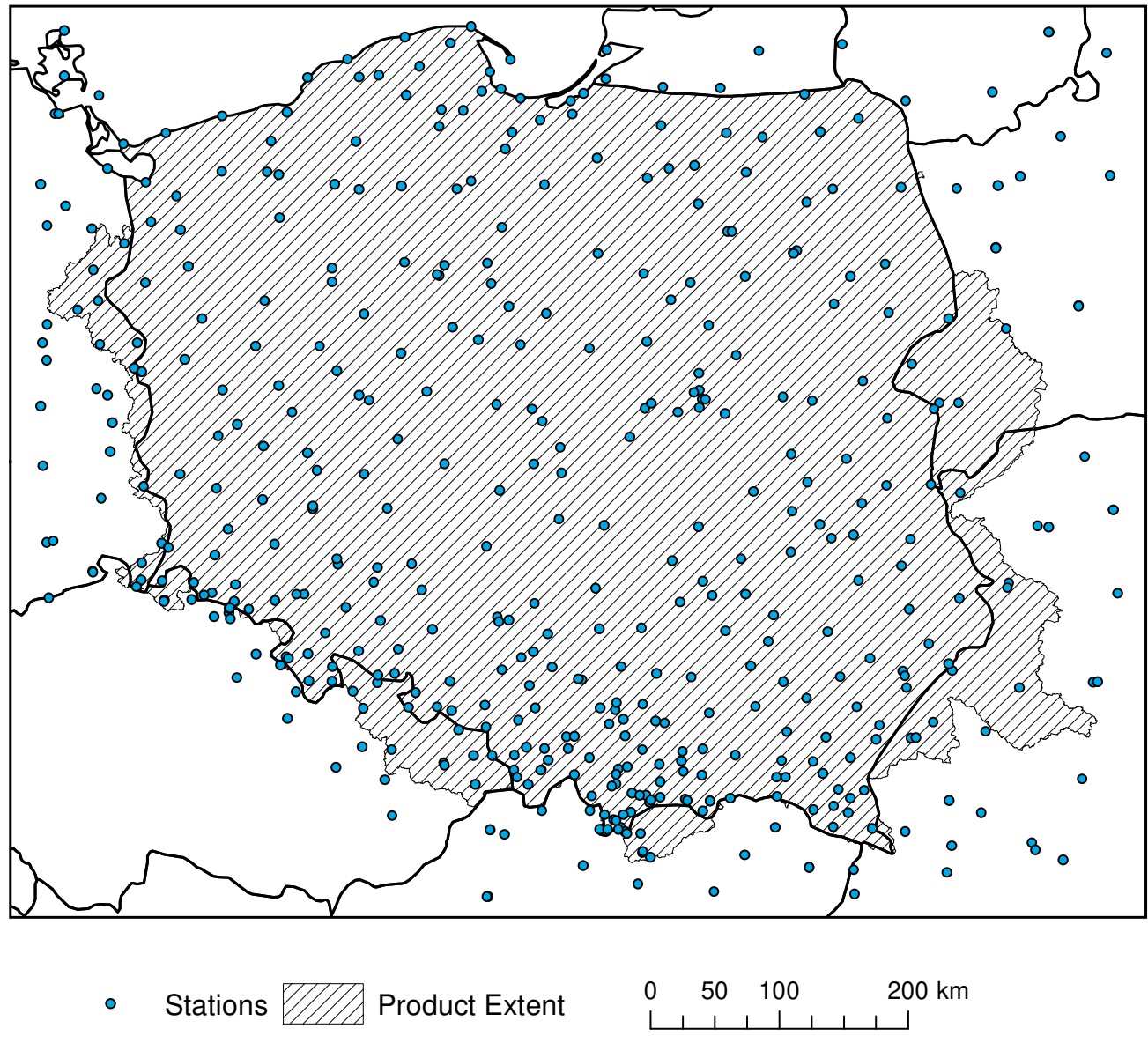

**Figure 9.** Spatial distribution of wind speed stations used for interpolation of the G2DC-PL+ product.

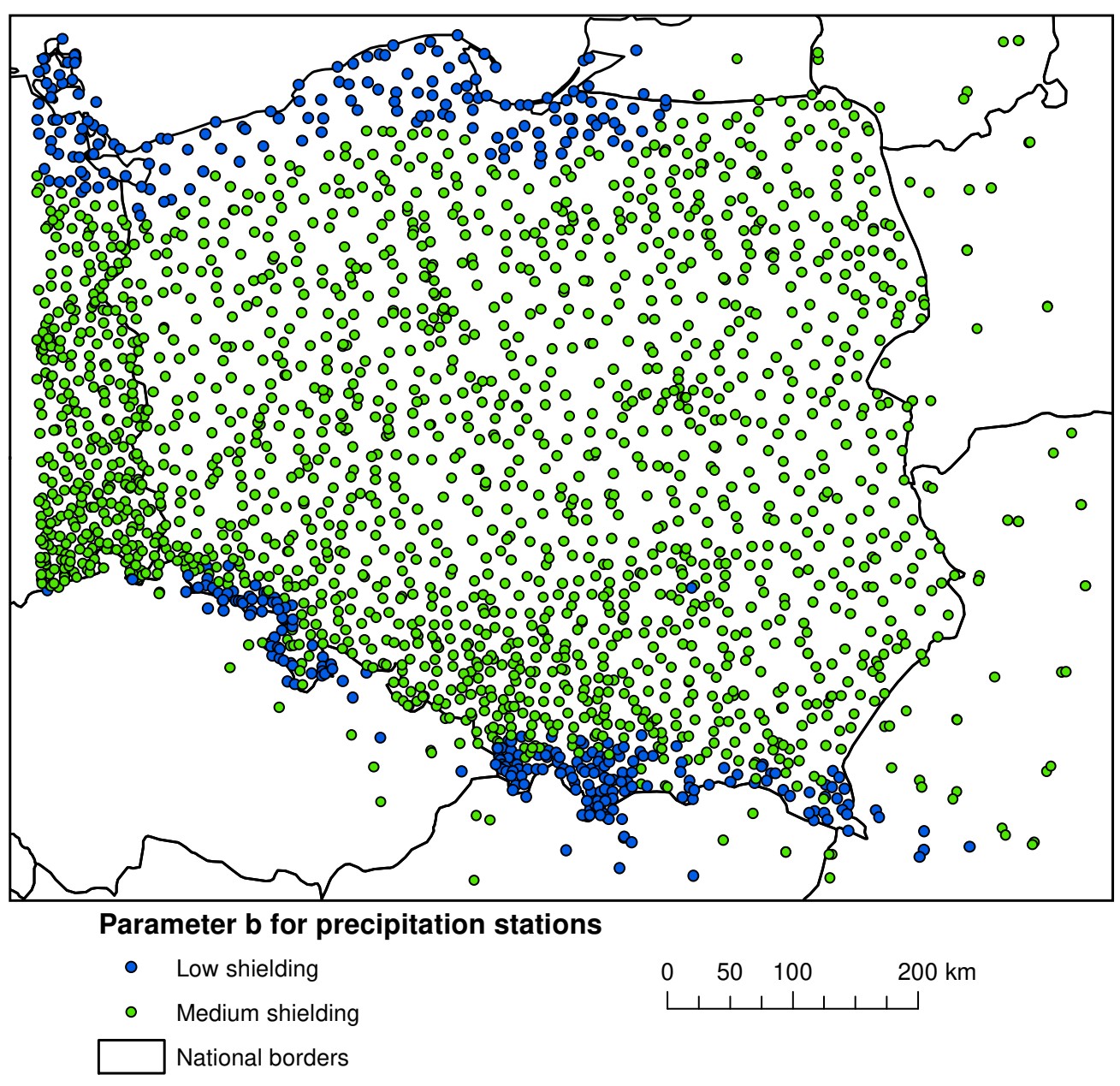

**Parameter b for precipitation stations**

- ● Low shielding
- ● Medium shielding
- ☐ National borders

0    50    100         200 km

**Figure 10.** The Richter (1995) parameter *b* value groups for different precipitation stations.

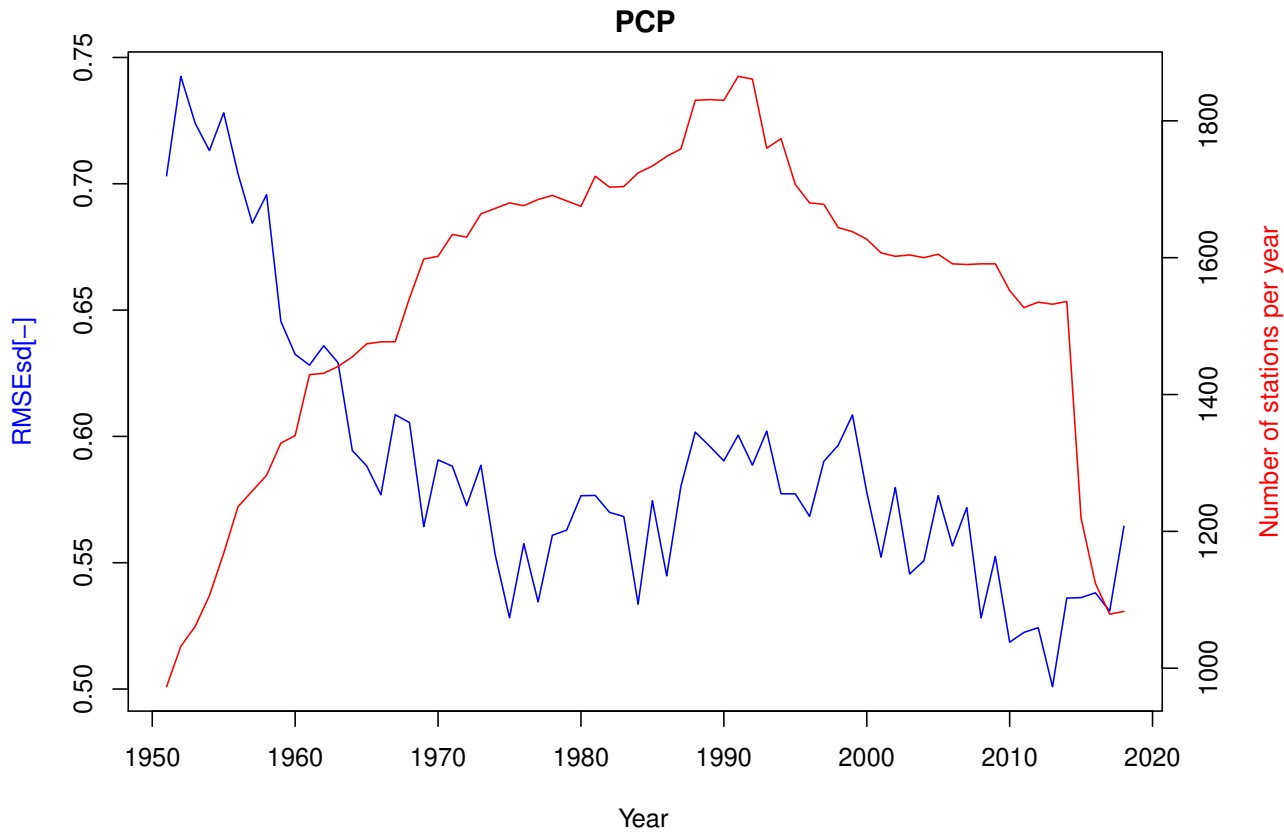

**Figure 11.** Annual RMSEsd median (blue) and number of available stations per year (red) for precipitation in the period 1951-2013. The daily results, also summarized in Table 1, were used for calculating the annual medians.

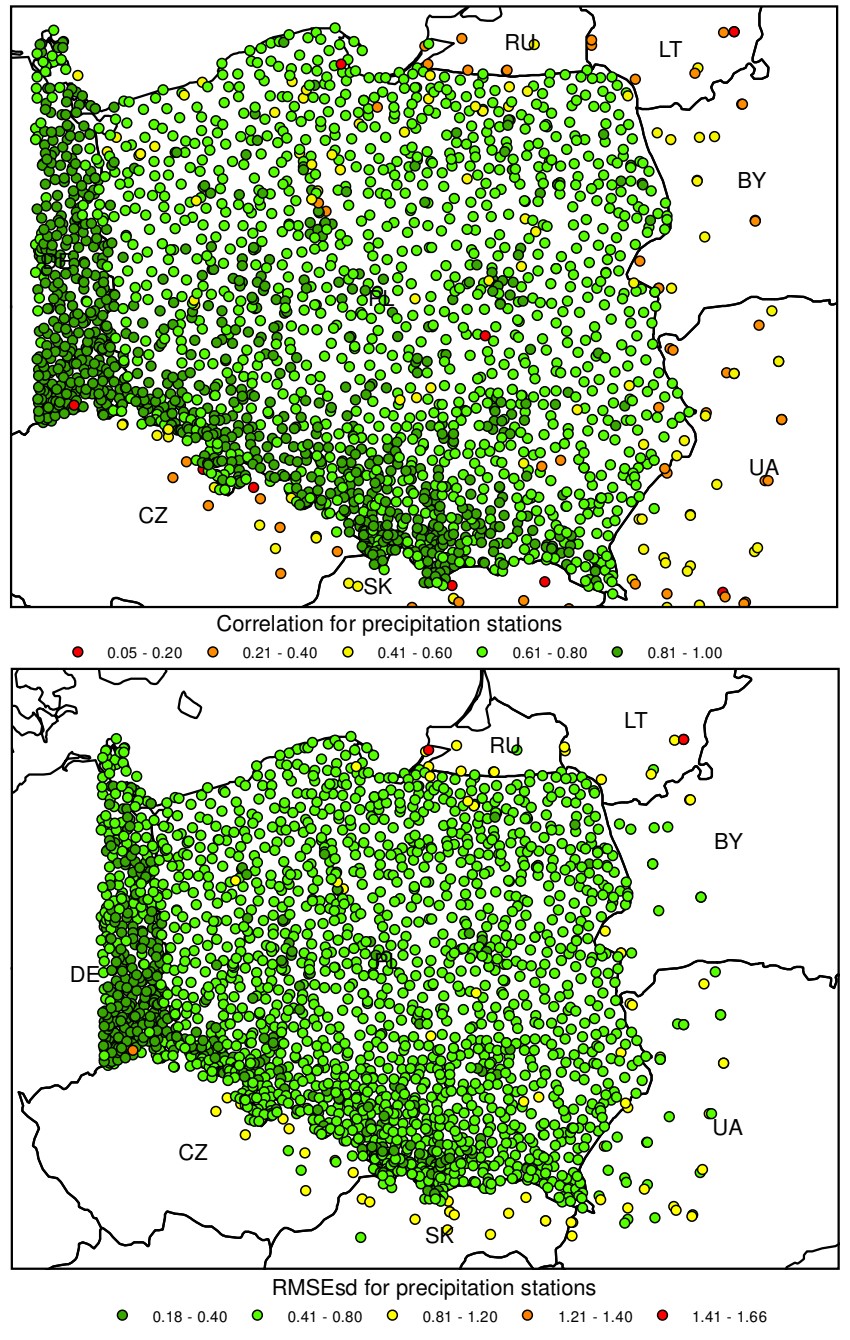

**Figure 12.** Precipitation $\rho$ (top) and RMSEsd (bottom) values calculated for stations in the period 1951-2019. National borders (black lines) are labelled with country codes.

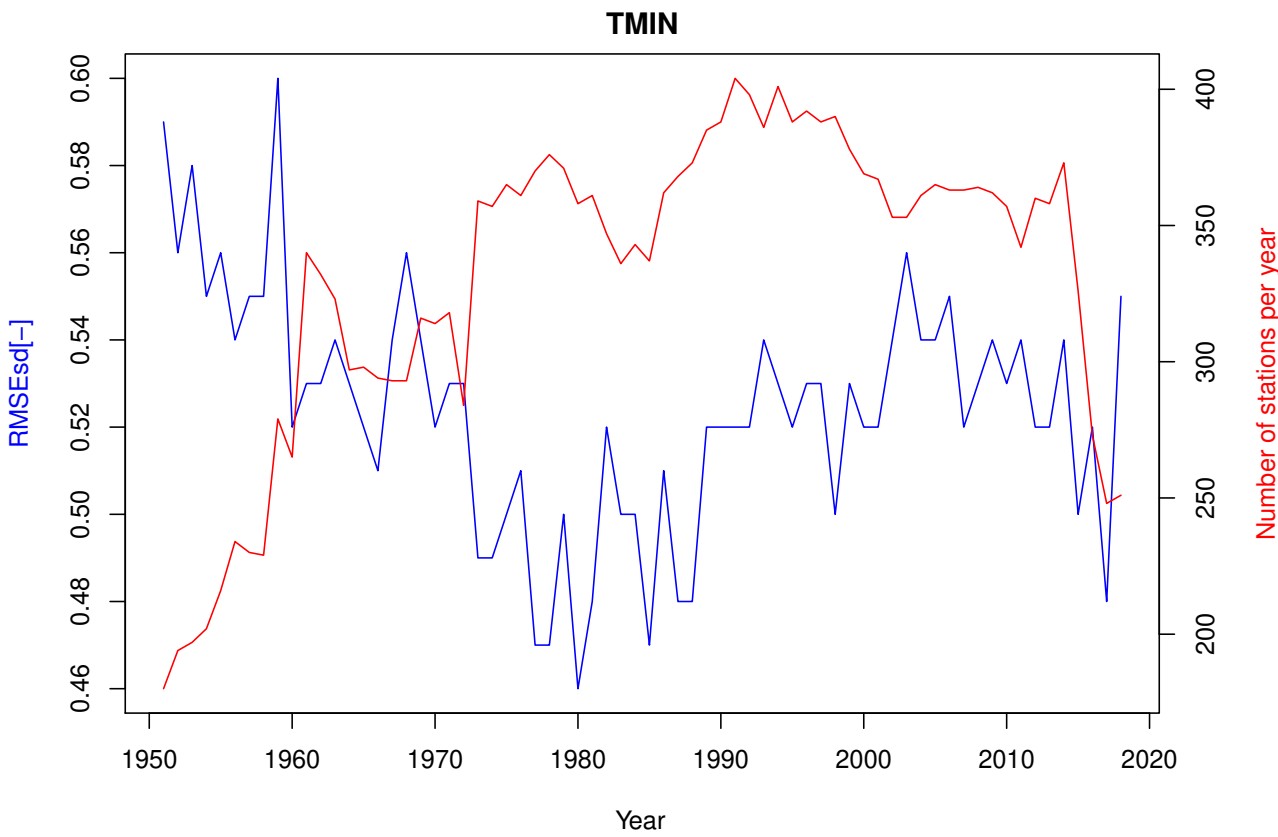

**Figure 13.** Annual RMSEsd median (blue) and number of available stations per year (red) for minimum temperature in the period 1951-2019. The daily results, also summarized in Table 1, were used for calculating the annual medians.

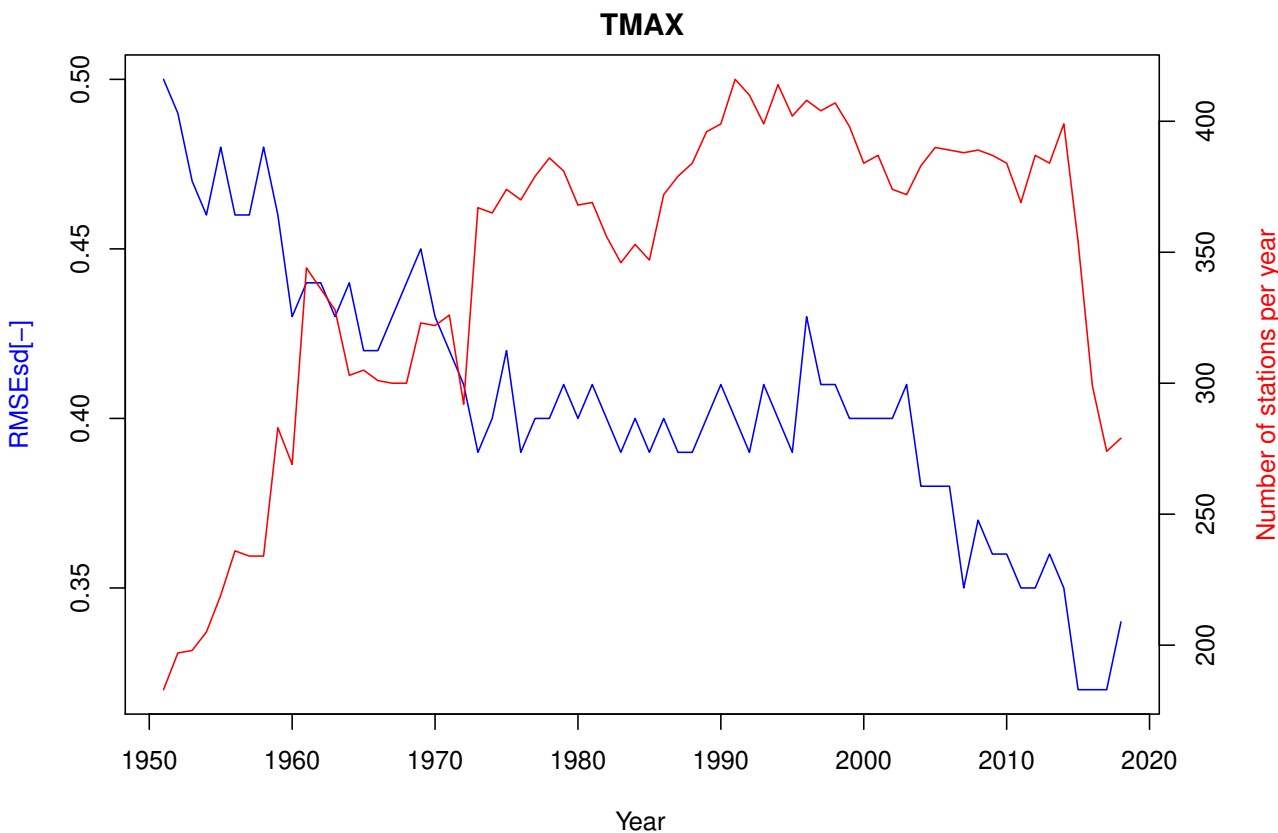

**Figure 14.** Annual RMSEsd median (blue) and number of available stations per year (red) for maximum temperature in the period 1951-2019. The daily results, also summarized in Table 1, were used for calculating the annual medians.

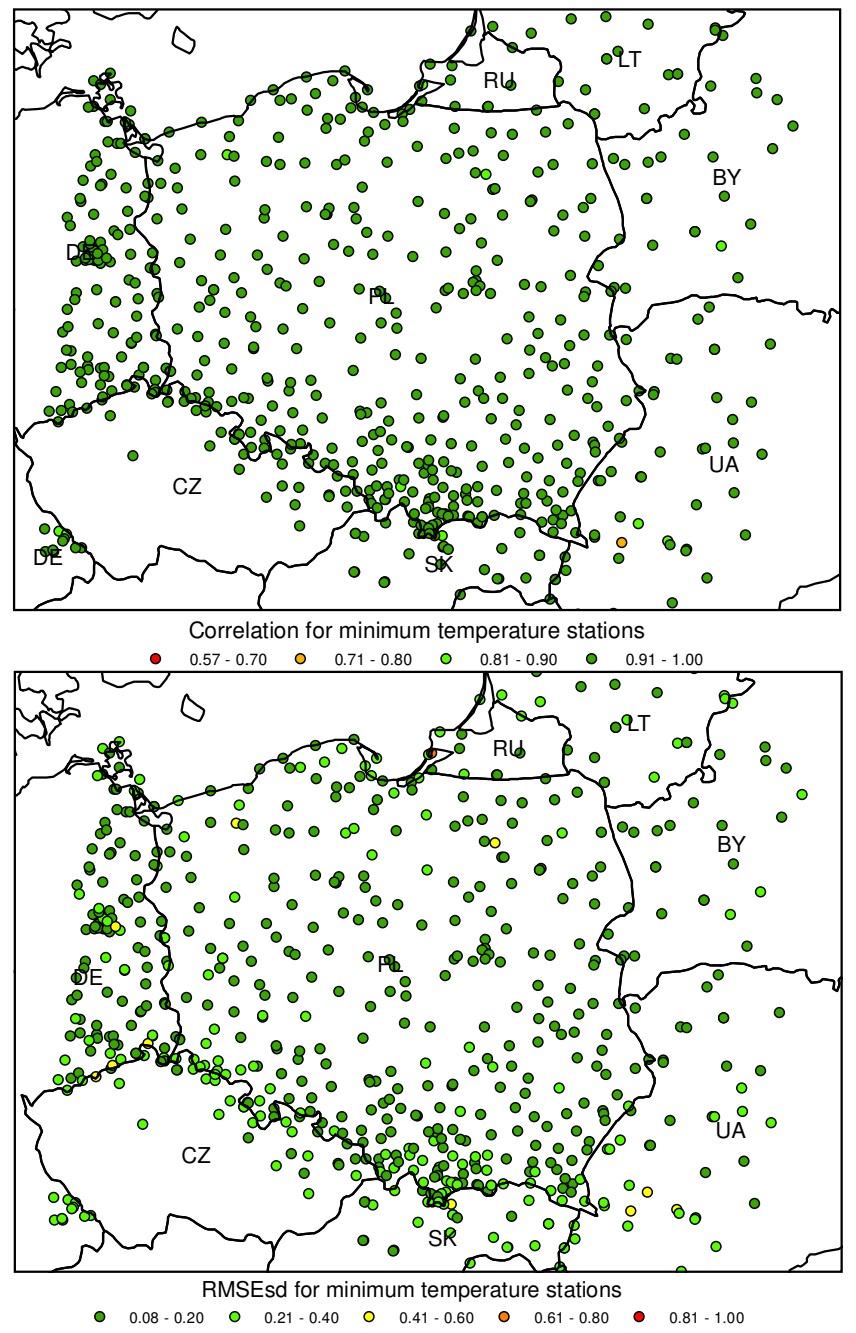

**Figure 15.** Minimum temperature $\rho$ (top) and RMSEsd (bottom) values calculated for stations in the period 1951-2019. National borders (black lines) are labelled with country codes.

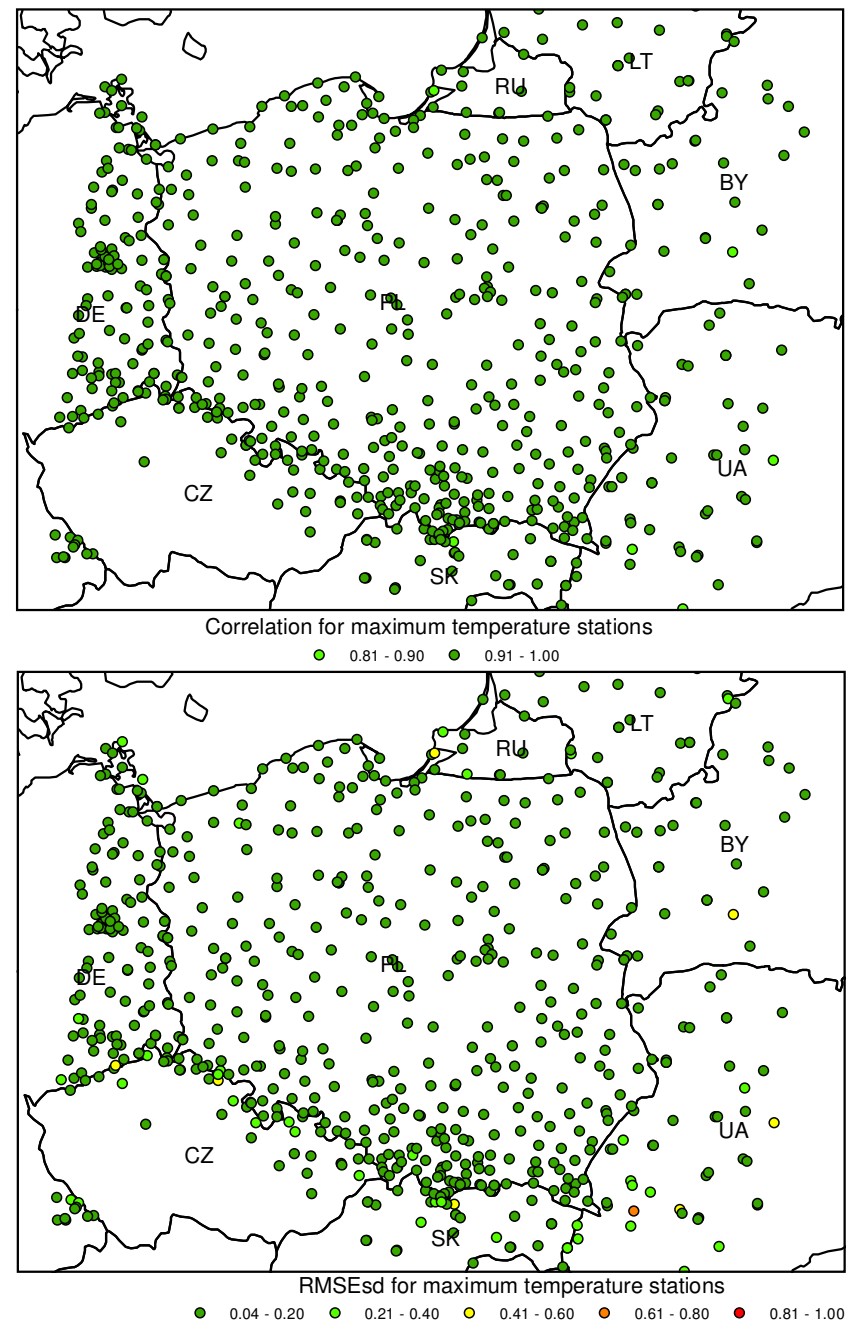

**Figure 16.** Maximum temperature $\rho$ (top) and RMSEsd (bottom) values calculated for stations in the period 1951-2019. National borders (black lines) are labelled with country codes.

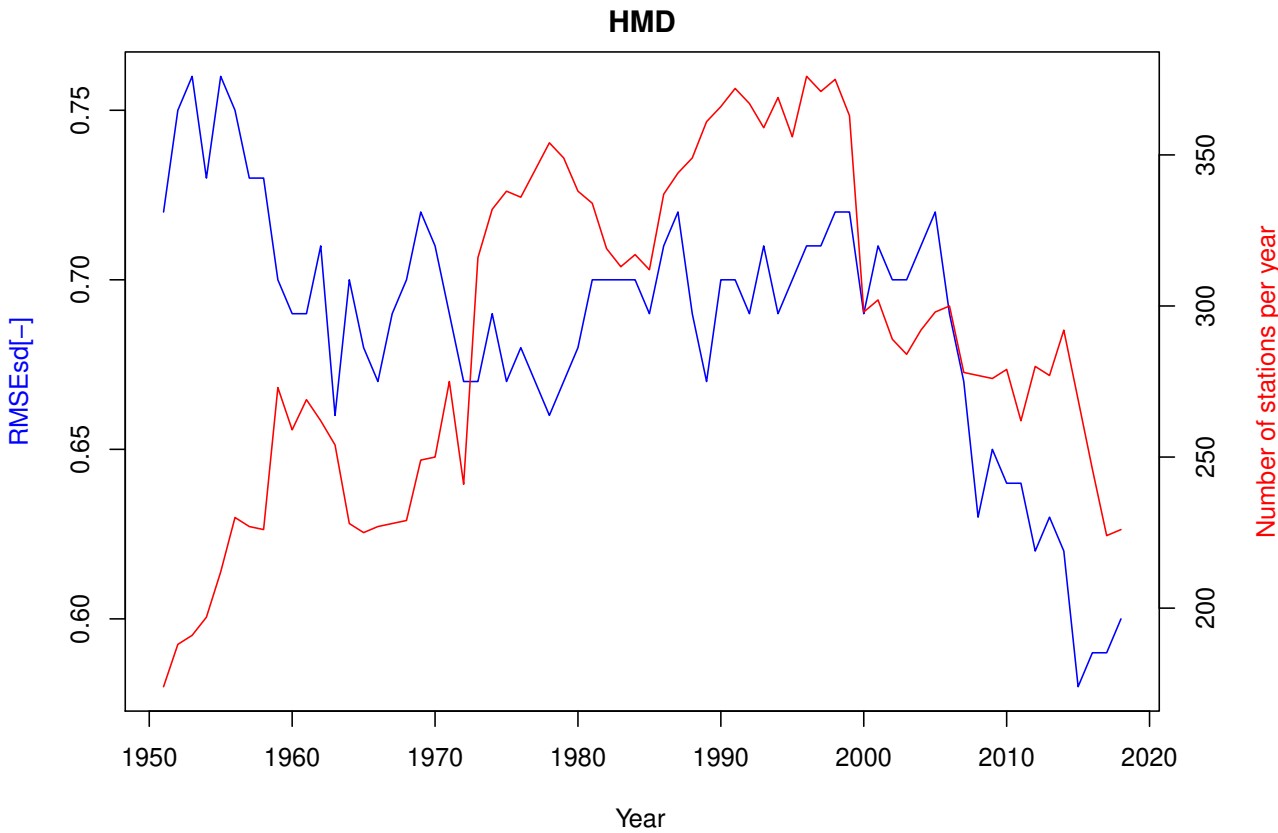

**Figure 17.** Annual RMSEsd median (blue) and number of available stations per year (red) for relative humidity in the period 1951-2019. The daily results, also summarized in Table 1, were used for calculating the annual medians.

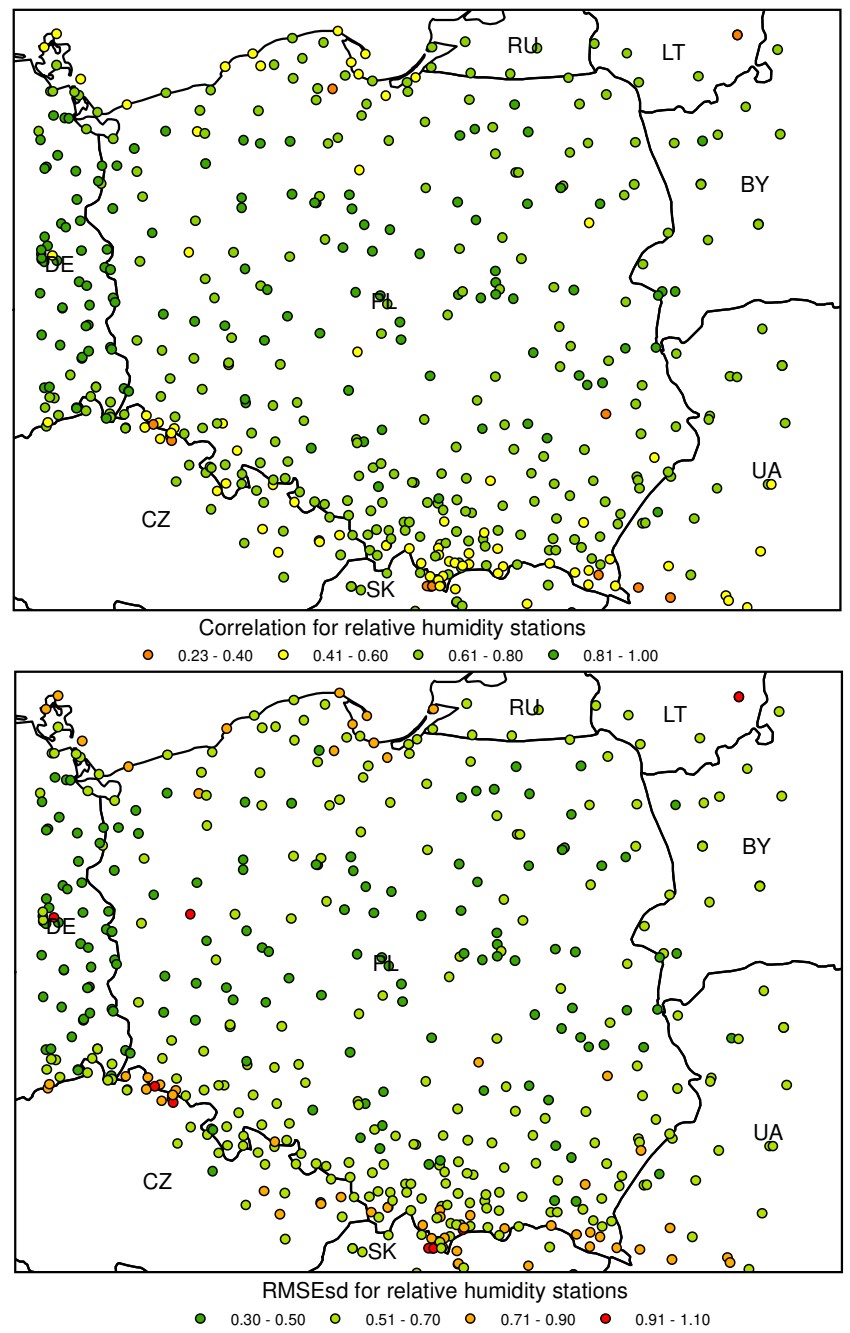

**Figure 18.** Relative humidity $\rho$ (top) and RMSEsd (bottom) values calculated for stations in the period 1951-2019. National borders (black lines) are labelled with country codes.

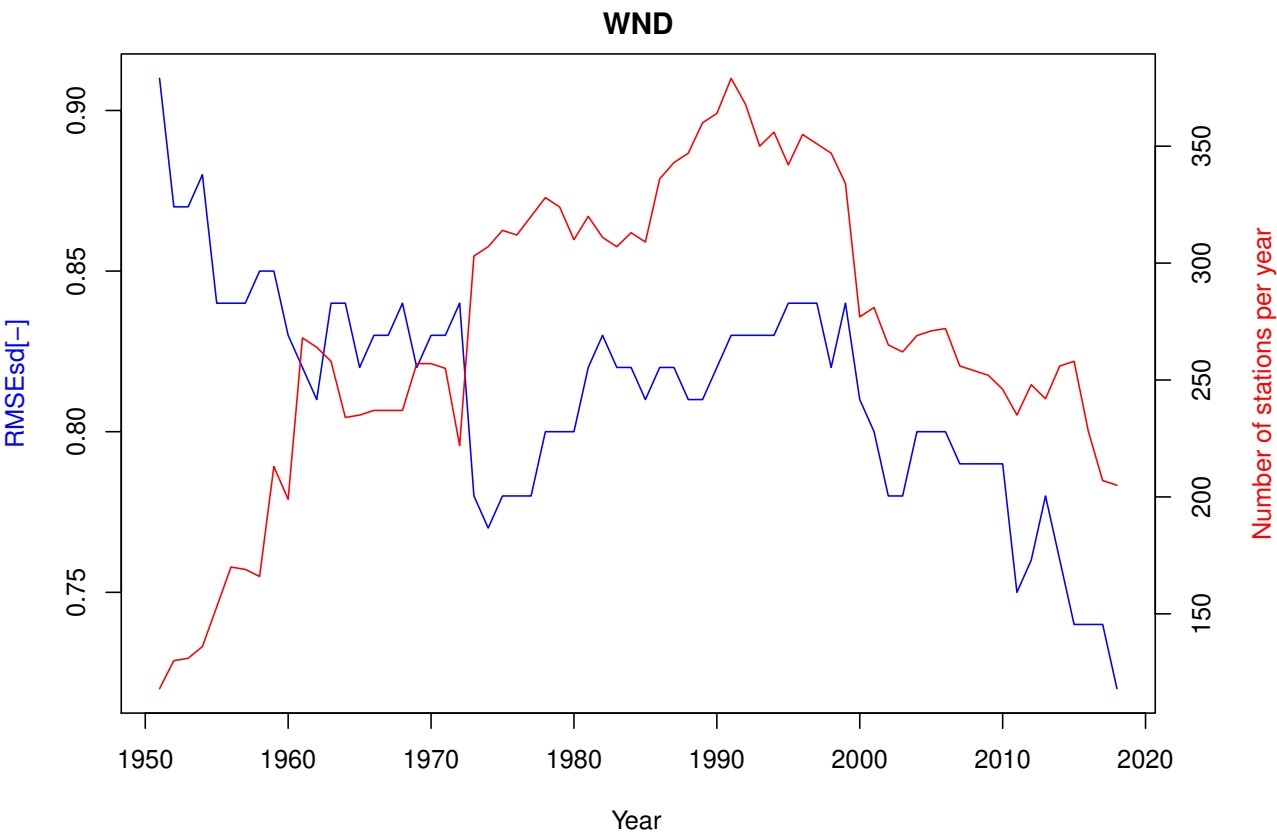

**Figure 19.** Annual RMSEsd median (blue) and number of available stations per year (red) for wind speed in the period 1951-2019. The daily results, also summarized in Table 1, were used for calculating the annual medians.

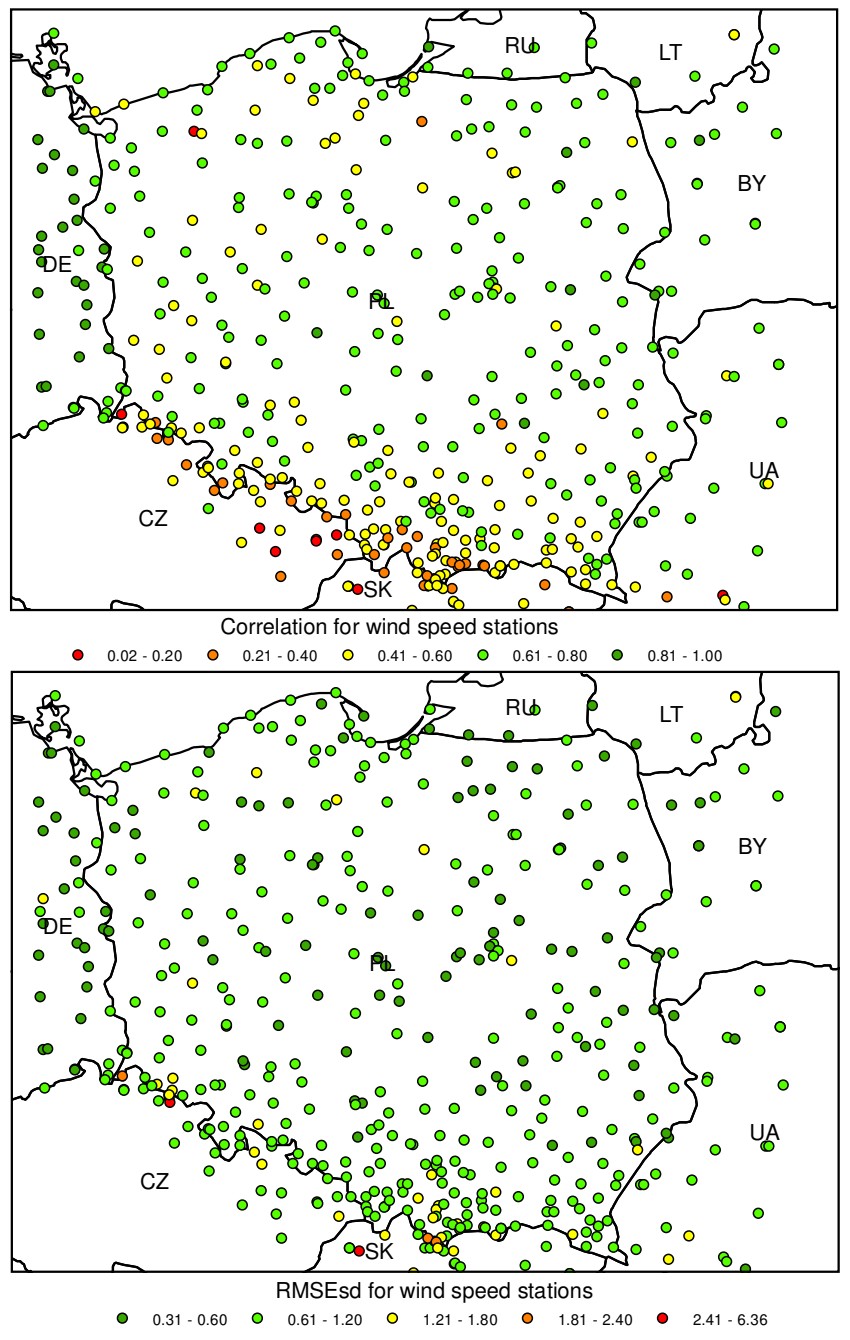

**Figure 20.** Wind speed $\rho$ (top) and RMSEsd (bottom) values calculated for stations in the period 1951-2019. National borders (black lines) are labelled with country codes.

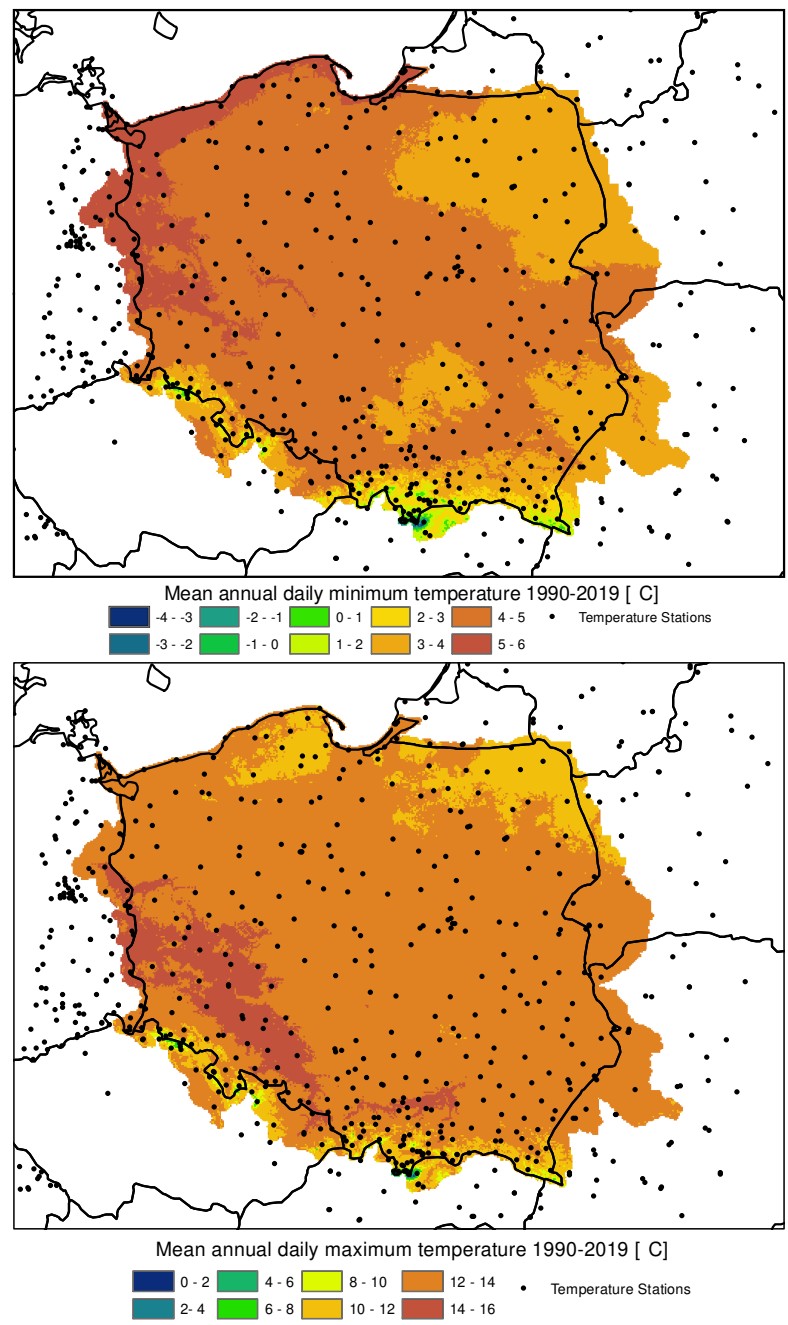

**Figure 21.** Mean annual daily minimum and maximum temperature in the time period 1990-2019: output from the G2DC-PL+ dataset.

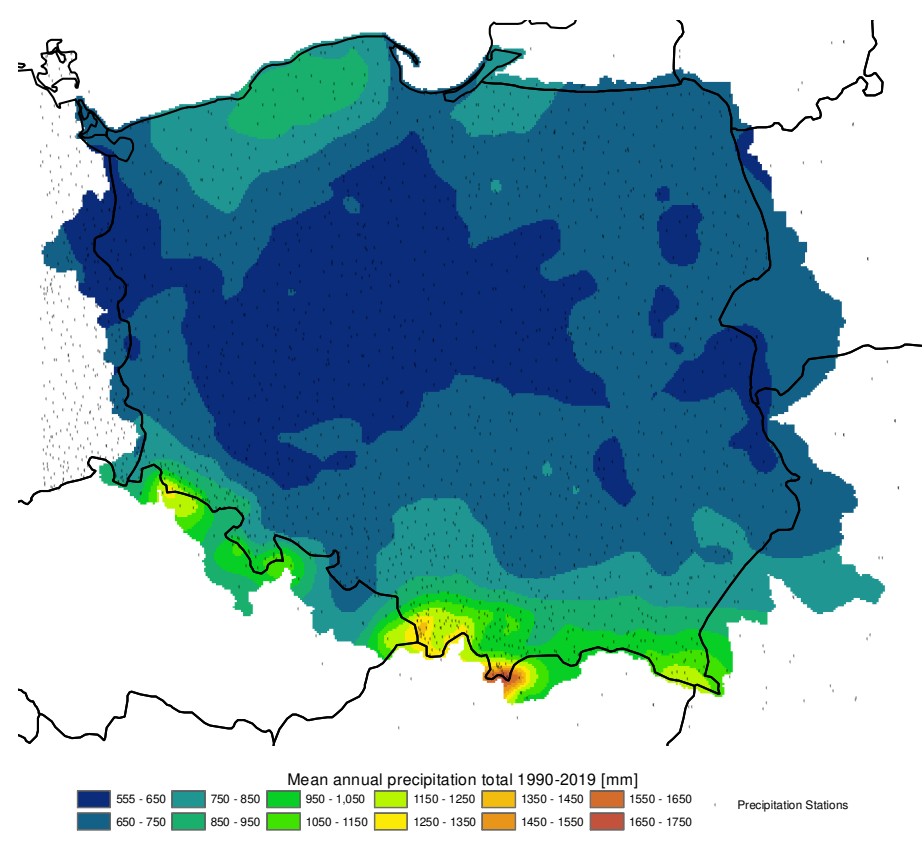

**Figure 22.** Mean annual precipitation in the time period 1990-2019: output from the G2DC-PL+ dataset.

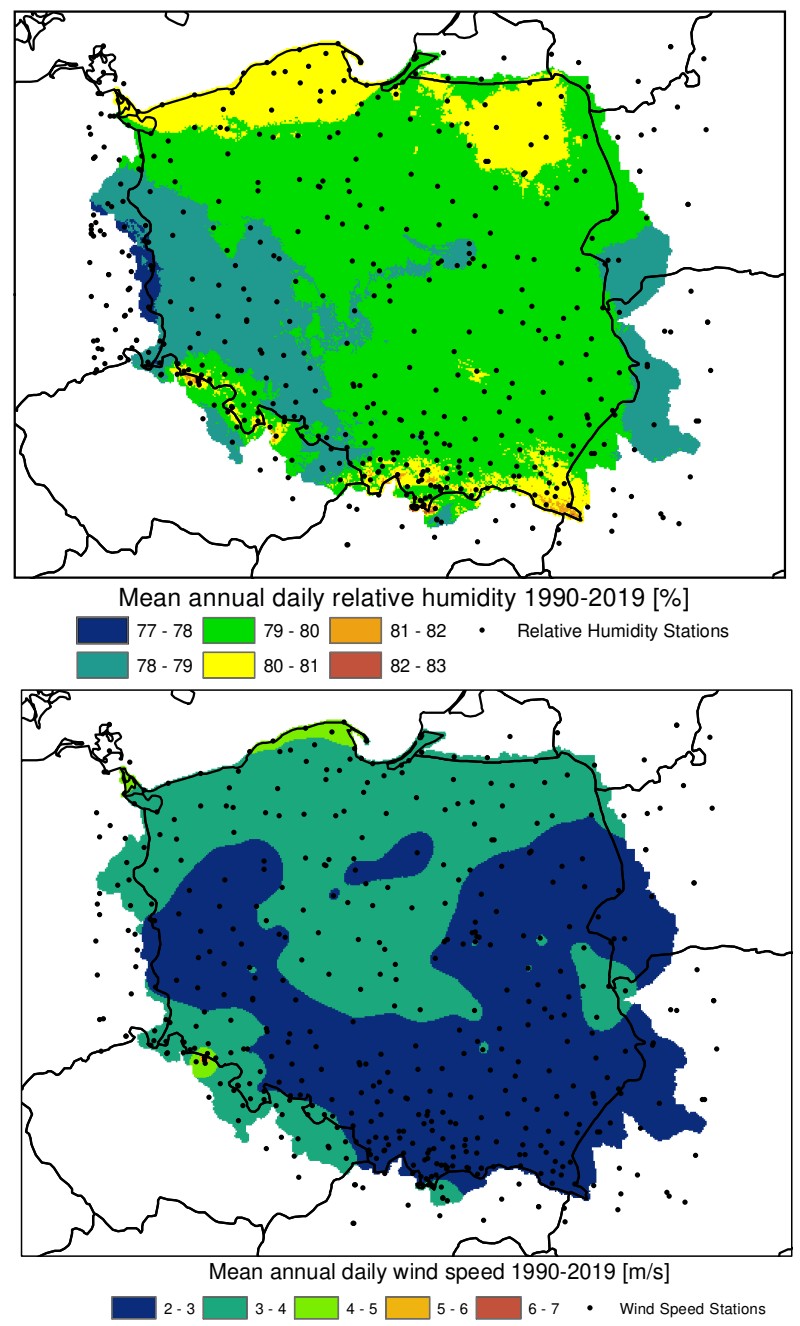

**Figure 23.** Mean annual relative humidity and wind speed in the time period 1990-2019: output from the G2DC-PL+ dataset.