# Peer review of "G2DC-PL+ A gridded 2 km daily climate dataset for the union of the Polish territory and the Vistula and Odra basins"

_Earth System Science Data, 2020_

## Referee Comment (RC1) · Anonymous Referee #1 · 13 Nov 2020

Review: "G2DC-PL+ A gridded 2 km daily climate dataset for the union of the Polish territory and the Vistula and Odra basins "

The manuscript describes a new version of very fine resolution analysis (named G2DC−PL+ with 2 km of horizontal grid spacing) that includes new atmospheric variables over the Polish territory and neighborhoods. The period of analysis is 1950-2019 and considers the variables precipitation, maximum and minimum temperature, and the new ones wind speed and relative humidity. The data sources to obtain the gridded analysis are clearly described as well the interpolation methods (basically variations of Kriging interpolation method). The correlation coefficients and the root mean square

error are used in the cross validation of the resultant gridded analysis. Comparisons are made with previous analysis at 5 km grid spacing and for the period 1950-2014. The authors highlight some improvements in the statistical indexes (correlation and root mean square error) for precipitation, while for air temperature (both minimum and maximum) this is not always the case. The manuscript describes how the gridded data are organized (variable names, time frequency, etc.), the formats (GeoTIFF and NeCDF), the address to download the data, etc. Overall, the G2DC−PL+ fine resolution gridded data have potential to contribute to hydrological and climate studies at local scale as well as will be useful to the validation of very high horizontal resolution climate simulations. I have just a few minor questions/suggestions for the authors. After this, my recommendation is to consider the manuscript suitable for publication. Minor comments: Please, to include in the first paragraph some information related to the relevance of fine scale gridded analysis to study local climate and its variability. L23 – What is [refs]? L35 – to correct the typo "(2019))" – "(2019)" L37 – What is the meaning of "GR4J"? L38 – remove "reported above" L43 – "modeling study for several medium-sized" should be "modeling studies for medium-sized" L48 – remove "ofrange" L60 – change "in (Berenzowski et al., 2016)" to "in Berenzowski et al. (2016)" L61 – change "in (Berezowski et al., 2019)" to "in Berezowski et al. (2019)" L61-63 – The information "As of July 2020, together . . . Science, respectively" is not relevant. Please, remove it. L69 – "spatial coverage " should be "domain" L72 – change "the dataset consists . . . data" should be "the new dataset consists of relative humidity and wind speed" L75 – "The six years temporal . . ." should be "The analysis cover the period 1951-2019. The six years extension (2014-2019) . . . " to make clear in this section (Temporal range) the total period considered. L77 – change "these years in the dataset" to "these years" and "help better constrain" to "help to constrain" L80 - correct the typo "in dataset" L87 – "much shorter . . ." to "shorter . . ." L119 – Was it necessary to interpolate the data in time or are the authors referring to "The time frequency for all variables was daily" of the data? I have similar doubt if in L132 "For each daily interpolation " is referring to "For daily data". Please, clarify in the manuscript. L144, L176 – change to "in Berezowski

et al. (2016)" L139 – remove "two functions" L139-140 – What is the meaning of [-]? L142 – remove "(precipitation and temperature)" L149 – remove " and RMSEsd" L151 – "14.2 % to" should be "14.2 % in Berezowski et al. (2016) to" L154 – "The results showed . . ." should be "We concluded . . ." L179 – remove "results " L185 – change "no prior data . . . as was the case of precipitation" to "no prior statistics . . . as occurred for precipitation" L219 – "In the present paper we have conducted . . ." should be "In addition, we present . . ." L184-L188 and L199-203 are practically identical (same words and sequence of ideas). Please, re-write. Figures Figures 12, 15, 16 and 20 have two very similar green colors, making it difficult to visualize the differences in the statistical indexes. Please, to improve these Figures.

---

## Referee Comment (RC2) · Joanna Wibig (Referee) · 20 Dec 2020

Interactive comment on G2DC-PL+ A gridded 2 km daily climate dataset for the union of the Polish territory and the Vistula and Odra basins by Piniewski et al.

The manuscript "G2DC-PL+ A gridded 2 km daily climate dataset for the union of the Polish territory and the Vistula and Odra basins" by Piniewski et al , submitted to Earth System Science Data presents a second, updated version of a high-resolution gridded daily precipitation, temperature, relative humidity and wind speed dataset as well as its description.

Overall, I assess very positively the main achievement of the paper – freely available dataset of high resolution daily data of containing key factors for calculating many climate parameters.

The manuscript is well organized and written quite clear. The reference list is complete. The methodology of interpolation method is based on the work by Berezowski et. al and only the differences between both approaches are described here. The description of the evaluation of interpolation errors is somewhat less clear and I would suggest a revision.

The interpolation errors were quantified using two functions: (1) Pearson's correlation coefficient ( [-]) and (2) root mean squared error normalized to standard deviation of the observed data [-]. The cross validation was conducted on both a temporal and spatial scale. On the temporal scale the errors were calculated for each day from all stations having data on this day and presented in the form of a descriptive statistics table. On the spatial scale the errors were calculated for each station from all of a station's available daily values. The number of records on the spatial scale calculation was equal to the number of meteorological stations used and results were presented in the form of maps.

Sounds good, but the devil is in the details. In this case it is a standard deviation. For temperature, is the standard deviation calculated for each julian day separately or for the whole record? Nothing in the equation indicates the first possibility. So the range of possibly values of daily minimum and maximum temperature is in the order of $50°C$, which means that the standard deviation is very large, as well as the interpolation error of the order of half of the standard deviation. Additionally, the distribution of daily minimum and maximum temperature from periods equal to the multiple of the year is bimodal. What is the statistical meaning of the standard deviation calculated from values from ther bimodal distribution. It is even worse in the case of precipitation. Distribution is far from normal and the standard deviation is a very poor measure of the spread of values.

The values of humidity and wind speed have much smaller dispersion and a less pro-pounced annual cycle, therefore the standard deviation describe their dispersion better and RMSEsd values are greater.

I have also some other minor comments:

The information on numer of citations (lines 61-63) is unnecassary and I suggest delet-ing it.

Line 77 should be years in instead of yearsin

Line 80 should be in dataset instead of indataset

Line 81 should be temperature instead of tempearture

Line 82 should be kriging instead of kirigging

In point 2 of subsection 2.4, the ending of the thought seems to be missing

In point 4 of subsection2.5, authors stated that "a map showing values of coefficient b representing the effect of wind exposition of the measurements site is presented in Figure 10", however in this Figure is only the division of stations into stations with low and medium shielding. It would be beneficial to provide some sort of breakdown criterion.

The differentiation of symbols denoting spatial and temporal values of the correlation coefficients and the RMSE would significantly facilitate the tracing of the text.

In the case of precipitation the relationship between the number of available stations and RMSE is intriguing. The increase in the number of stations in the initial period was accompanied by the decline of the RMSE. It is clear. However, in the second part of the analysed record, the decrease in the number of stations was not accompanied by an increase in the RMSE. It is surprising. The authors only stated the fact without comment. And it would be very useful.

When writing about temperature interpolation errors the authors only stated that the conclusion that kriging errors for temperature are not dependent on the density of the observation network seems to hold true. They did not comment this fact. Some discussion on the impact of spatial correlation of variables on the relationship between station density and RMSE would be beneficial.

In my opinion the lack of discussion is the main drawback of the paper.

---

## Referee Comment (RC3) · Anonymous Referee #3 · 7 Jan 2021

Review of

G2DC-PL+ A gridded 2 km daily climate dataset for the union of the Polish territory and the Vistula and Odra basins by Mikołaj Piniewski, Mateusz Szcześniak, Ignacy Kardel, Somsubhra Chattopadhyay, and Tomasz Berezowski. Submitted to Earth System Science Data

Summary

The manuscript evaluates the new freely available gridded observation dataset (kriging method) for Poland and the Vistula and Odra basins, G2DC-PL+, at 2 km resolution covering 1951-2019, which is an updated and extended version of the 5 km 1951-2013

[Figure]

CPLFD-GDPT5 data set from the previous CHASE-PL project. Daily precipitation and min and max temperatures are extended to include relative humidity and wind speed, use of more observation stations (approx. doubled for temperature and precipitation) due to new freely available IMGW−PIB climate data and from other sources. Two applications of such datasets are hydrological modelling and bias correction of climate model output. The new data are presented with statistical measures (e.g. from cross-validation) with comparison with the previous dataset of temperature and precipitation.

General comments

In my opinion, the manuscript is a result of solid work with a clear presentation and sufficient level of details. However, although the title of the manuscript is "A gridded 2 km ...", there is not a single map of the gridded products using classical presentation with isolines or colour shadings in the main manuscripts, but reference to supplementary material in section 4 Consistency with climatic data. In this way there is no visualization for the reader of the spatial details in the final gridded product which can be compared to e.g. station density or terrain. I would not recommend increasing the total number of figures, but would like to see some presentation of e.g. mean values. A combination of the present point station presentation on top, using isolines and/or colour shading as background of mean values, would certainly be possible for temperature, wind speed and relative humidity, while more difficult for precipitation due to the high station density. Additionally, the manuscript should include some discussion of the useful spatial scales of the gridded product (2 km scale for all parameters) in view of the much lower station density for most parameters (except precipitation), see specific comments.

Specific comments

As far as I can see, some improvement of the English grammar is needed in a few places.

Space is missing between 2 words at line number 10, 77, 88, 186.

Corrections for missing or extra space before or after "," needed (text search for comma).

Line 23 [refs] needs to be specified.

Line 33 applicatons -> applications

Line 47 remove "ofrange"

Line 65 remove comma+space in "interpolation, ."

Line 77 "will help better constrain hydrological models" rephrase?

Line 82 "kirigging" correct.

Line 91 "evap-otransipration" correct.

Line 120 "under-catch" -> "undercatch" for consistent naming in the manuscript..

Line 124 "(Berezowski et al., 2016)" -> "Berezowski et al. (2016)" ?

Line 139-140 I guess that "[-]" means without unit, should in my opinion be modified or removed to avoid confusion for the reader.

2.4 Number of stations. This section or the concluding section should in my opinion be supplemented with some discussion of the useful spatial accuracy of the final 2 km data in relation to the station density for the different parameters.

---

## Author Comment (AC1) · 26 Jan 2021

We would like to thank all the Reviewers for posting comments that will help improve our manuscript. We provided point-by-point responses (in blue) to individual comments of each Reviewer below (in black).

**Reviewer #1:**

The manuscript describes a new version of very fine resolution analysis (named G2DC–PL+ with 2 km of horizontal grid spacing) that includes new atmospheric variables over the Polish territory and neighborhoods. The period of analysis is 1950-2019 and considers the variables precipitation, maximum and minimum temperature, and the new ones wind speed and relative humidity. The data sources to obtain the gridded analysis are clearly described as well the interpolation methods (basically variations of Kriging interpolation method). The correlation coefficients and the root mean square error are used in the cross validation of the resultant gridded analysis. Comparisons are made with previous analysis at 5 km grid spacing and for the period 1950-2014. The authors highlight some improvements in the statistical indexes (correlation and root mean square error) for precipitation, while for air temperature (both minimum and maximum) this is not always the case. The manuscript describes how the gridded data are organized (variable names, time frequency, etc.), the formats (GeoTIFF and NeCDF), the address to download the data, etc. Overall, the G2DC–PL+ fine resolution gridded data have potential to contribute to hydrological and climate studies at local scale as well as will be useful to the validation of very high horizontal resolution climate simulations. I have just a few minor questions/suggestions for the authors. After this, my recommendation is to consider the manuscript suitable for publication.

**Response**: We thank the reviewer for his time, kind words and have now addressed all the suggested minor changes in the manuscript.

Minor comments

1) Please, to include in the first paragraph some information related to the relevance of fine scale gridded analysis to study local climate and its variability.

**Response**: Excellent suggestion, we have now added relevant information regarding importance of fine scale gridded analysis to study local climate and variability.

The following lines were now added to the manuscript:

Climate change across the landscape has significant spatiotemporal variations which are often not uniform or consistent (Hayhoe et al., 2007). Spatial heterogeneity in the distribution of earth surface features including physical variables such as land cover, soil moisture as well as landscape properties such as slope, elevation interact with large scale climate which in turn determines microscale climate (Dobrowski et al., 2009). Herein lies the significance of fine scale gridded analysis to study local climate and variability.

**2) L23 – What is [refs]?**

**Response:** Proper reference is now provided.

3) L35 – to correct the typo "(2019))" – "(2019)"

Response: Done.

4) L37 – What is the meaning of "GR4J"?

Response: We added an explanation: in French, modèle du Génie Rural à 4 paramètres Journalier

5) L38 - remove "reported above"

Response: Done.

6) "L43 - "modeling study for several mediumsized" should be "modeling studies for medium-sized"

Response: Done.

7) L48 - remove "ofrange"

Response: Done.

8) L60 - change "in (Berenzowski et al., 2016)" to "in Berenzowski et al. (2016)"

Response: Done.

9) L61 - change "in (Berezowski et al., 2019)" to "in Berezowski et al. (2019)"

**Response:** As suggested by reviewer #2, we have now deleted the information on lines 61-63.

10) L61-63 – The information "As of July 2020, together . . . Science, respectively" is not relevant. Please, remove it.

Response: Agreed and these lines are now deleted from the manuscript.

11) L69 - "spatial coverage " should be "domain"

Response: Done.

12) L72 – change "the dataset consists . . . data" should be "the new dataset consists of relative humidity and wind speed"

Response: Done.

13) L75 – "The six years temporal . . ." should be "The analysis cover the period 1951-2019. The six years extension (2014-2019) . . . " to make clear in this section (Temporal range) the total period considered.

Response: Sentence revised.

14) L77 – change "these years in the dataset" to "these years" and "help better constrain" to "help to constrain"

Response: Done.

15) L80 - correct the typo "in dataset"

Response: Done.

16) L87 – "much shorter . . ." to "shorter . . ."

Response: Done.

17) "L119 – Was it necessary to interpolate the data in time or are the authors referring to "The time frequency for all variables was daily" of the data? I have similar doubt if in L132 "For each daily interpolation " is referring to "For daily data". Please, clarify in the manuscript.

**Response:** We agree that this phrasing was not clear. We revised it following the reviewer's suggestion.

18) L144, L176 - change to "in Berezowski et al. (2016)"

Response: Done.

19) L139 - remove "two functions"

Response: Done.

20) L139-140 – What is the meaning of [-]?

Response: Following a suggestion of Reviewer #3, we removed it from the text.

21) L142 - remove "(precipitation and temperature)"

Response: Done.

22) L149 - remove " and RMSEsd"

Response: Done.

23)" L151 - "14.2 % to" should be "14.2 % in Berezowski et al. (2016) to"

Response: Done.

24) L154 – "The results showed . . ." should be "We concluded . . ."

Response: Sentence revised.

25) L179 – remove "results

Response: Done.

26) "L185 – change "no prior data . . . as was the case of precipitation" to "no prior statistics . . . as occurred for precipitation"

**Response: Changed.**

27) "L219 – "In the present paper we have conducted . . ." should be "In addition, we present . . ."

**Response:** Changed

28) L184-L188 and L199-203 are practically identical (same words and sequence of ideas). Please, re-write

**Response:** Thank you for pointing this out, we have reworded these sentences.

29) Figures 12, 15, 16 and 20 have two very similar green colors, making it difficult to visualize the differences in the statistical indexes. Please, to improve these Figures

**Response:** We agree with this suggestion and have now updated these figures with a different color scheme to improve the clarity in the message.

**Reviewer #2:**

The manuscript "G2DC-PL+ A gridded 2 km daily climate dataset for the union of the Polish territory and the Vistula and Odra basins" by Piniewski et al , submitted to Earth System Science Data presents a second, updated version of a high-resolution gridded daily precipitation, temperature, relative humidity and wind speed dataset as well as its description. Overall, I assess very positively the main achievement of the paper

- freely available dataset of high resolution daily data of containing key factors for calculating many climate parameters. The manuscript is well organized and written quite clear. The reference list is complete. The methodology of interpolation method is based on the work by Berezowski et. al and only the differences between both approaches are described here.

**Response: We thank Prof. Joanna Wibig for her positive opinion about the manuscript.**

The description of the evaluation of interpolation errors is somewhat less clear and I would suggest a revision. The interpolation errors were quantified using two functions: (1) Pearson's correlation coefficient ([-]) and (2) root mean squared error normalized to standard deviation of the observed data [-]. The cross validation was conducted on both a temporal and spatial scale. On the temporal scale the errors were calculated for each day from all stations having data on this day and presented in the form of a descriptive statistics table. On the spatial scale the errors were calculated for each station from all of a station's available daily values. The number of records on the spatial scale calculation was equal to the number of meteorological stations used and results were presented in the form of maps.

Sounds good, but the devil is in the details. In this case it is a standard deviation. For temperature, is the standard deviation calculated for each julian day separately or for the whole record? Nothing in the equation indicates the first possibility. So the range of possibly values of daily minimum and maximum temperature is in the order of 50°C, which means that the standard deviation is very large, as well as the interpolation error of the order of half of the standard deviation. Additionally, the distribution of daily minimum and maximum temperature from periods equal to the multiple of the year is bimodal. What is the statistical meaning of the standard deviation calculated from values from ther bimodal distribution. It is even worse in the case of precipitation. Distribution is far from normal and the standard deviation is a very poor measure of the spread of values.

The values of humidity and wind speed have much smaller dispersion and a less propounced annual cycle, therefore the standard deviation describe their dispersion better and RMSEsd values are greater.

**Response:** We are very thankful for pointing out the issue related to using a standardised version of the root mean squared error as a cross-validation indicator.

We should have clarified that for all variables standard deviation used in calculation of RMSEsd performance metrics was calculated for each Julian day separately. We thus believe that the issue related to the non-normal distribution of daily minimum and maximum temperature and precipitation variables is not really affecting the interpretation of RMSEsd values. Further, for comparability of our results we followed Belo-Pereira et al. (2011) in using RMSEsd as a measure of interpolation error.

However, we are aware that this kind of standardization is not very common and 'raw' RMSE values have been reported more frequently in scientific publications evaluating interpolation errors. In the revised manuscript we also provided the values of RMSE in Table 1. Median RMSE values are: 1.64 mm, 1.33 °C, 1.11 °C, 0.06 and 1.56 m/s, for precipitation, minimum temperature, maximum temperature, relative humidity and wind speed, respectively. We also analysed the spatial pattern in RMSE and compared it with a spatial pattern in RMSEsd for each variable. Both patterns were largely similar, the main difference was, not surprisingly, that RMSE values were usually relatively larger in areas where the magnitude of a given variable was higher. For example, RMSEsd was of similar range in the mountains and lowlands, whereas RMSE was clearly higher in the mountains than in the lowlands.

Information about the values of RMSE as well as its relation to RMSEsd was added to respective sections 3.1-3.5.

Minor comments

1) The information on number of citations (lines 61-63) is unnecessary and I suggest deleting it.

Response: Done.

2) Line 77 should be years in instead of yearsin

Response: Done.

3) Line 80 should be in dataset instead of indataset

Response: Done.

4) Line 81 should be temperature instead of tempearture

Response: Done.

5) Line 82 should be kriging instead of kirigging

Response: Done.

6) In point 2 of subsection 2.4, the ending of the thought seems to be missing

**Response:** There is no point 2 in subsection 2.4 so we assume that the Referee had subsection 2.5 in mind here. Point 2 of this subsection reads: "All organisations from which we have compiled the data conduct quality control check for raw data before making them publicly available." This sentence only conveys the message that all the raw data that we used underwent standard quality control checks by respective organizations.

7) In point 4 of subsection 2.5, authors stated that "a map showing values of coefficient b representing the effect of wind exposition of the measurements site is presented in Figure 10", however in this Figure is only the division of stations into stations with low and medium shielding. It would be beneficial to provide some sort of breakdown criterion.

**Response:** We followed the same, simplified criteria for dividing stations into those with 'low' and 'medium' shielding as Berezowski et al. (2016). Stations located above 400 m a.s.l. and those lying within a 40 km buffer from the coast were assigned to a 'medium' shielding category.

The differentiation of symbols denoting spatial and temporal values of the correlation coefficients and the RMSE would significantly facilitate the tracing of the text.

**Response:** This is a great suggestion, we followed it in the revised version.

In the case of precipitation the relationship between the number of available stations and RMSE is intriguing. The increase in the number of stations in the initial period was accompanied by the decline of the RMSE. It is clear. However, in the second part of the analysed record, the decrease in the number of stations was not accompanied by an increase in the RMSE. It is surprising. The authors only stated the fact without comment. And it would be very useful.

**Response:** Thank you for that observation. We find that this issue is quite complex; for example, an increase of the number of stations was very high until ~1965, but between 1965 and 1990 the rise was lower, although stable. This stable increase was not accompanied, though, by a further decrease in RMSEsd. In contrast, RMSEsd was fluctuating in this period in a rather narrow range 0.53 - 0.61. We do not have

any reliable data on how measurement quality changed over time, and thus are unable to confirm a potentially relevant hypothesis that it was an important factor affecting the evolution of interpolation error.

When writing about temperature interpolation errors the authors only stated that the conclusion that kriging errors for temperature are not dependent on the density of the observation network seems to hold true. They did not comment this fact. Some discussion on the impact of spatial correlation of variables on the relationship between station density and RMSE would be beneficial. In my opinion the lack of discussion is the main drawback of the paper.

**Response:** We concur with the reviewer's view that spatial correlation is indeed an important parameter that could play a role in station density and RMSE. However, due to a large volume of this dataset (5 variables, daily data for 69 years and the number of stations per variable ranging from 150 to 1800) calculation of auto-correlation would require a high computational time that would largely exceed the time given to submit the revision. Yet, in order to better understand the relationship between RMSEsd and station density we performed additional analyses. For each variable, we calculated Pearson's correlation coefficient between RMSEsd and station density. The latter was calculated assuming 50 km radius around each station and weight coefficients inversely proportional to the distance from a station. RMSEsd values were restricted to the 10-year periods with the highest data availability for each variable (usually 1990s) in order to minimize the effect of data availability fluctuations. The results showed that only for precipitation there is a moderate relationship between station density and RMSEsd (Pearson's R = 0.5, p < 0.0001, N = 1938). For all other variables it appeared that a strong positive correlation between station density and elevation potentially masks the effect of station density on RMSEsd. These results are also supported by those obtained by Berndt and Haberlandt (2018) who concluded that the influence of station density on the performance of different interpolation techniques is lower than that of temporal resolution and spatial variability.

**Reviewer #3:**

The manuscript evaluates the new freely available gridded observation dataset (kriging method) for Poland and the Vistula and Odra basins, G2DC-PL+, at 2 km resolution covering 1951-2019, which is an updated and extended version of the 5 km 1951-2013 CPLFD-GDPT5 data set from the previous CHASE-PL project. Daily precipitation and min and max temperatures are extended to include relative humidity and wind speed, use of more observation stations (approx. doubled for temperature and precipitation) due to new freely available IMGW–PIB climate data and from other sources. Two applications of such datasets are hydrological modelling and bias correction of climate model output. The new data are presented with statistical measures (e.g. from cross validation) with comparison with the previous dataset of temperature and precipitation.

**General comments**

In my opinion, the manuscript is a result of solid work with a clear presentation and sufficient level of details. However, although the title of the manuscript is "A gridded 2 km ...", there is not a single map of the gridded products using classical presentation with isolines or colour shadings in the main manuscripts, but reference to supplementary material in section 4 Consistency with climatic data. In this way there is no visualization for the reader of the spatial details in the final gridded product which can be compared to e.g. station density or terrain. I would not recommend increasing the total number of figures, but would like to see some presentation of e.g. mean values. A combination of the present point station presentation on top, using isolines and/or colour shading as background of mean values, would certainly be possible for

temperature, wind speed and relative humidity, while more difficult for precipitation due to the high station density.

**Response:** The reviewer's time and kind words are appreciated. We agree to this suggestion and have now included three new figures showing the spatial distribution of temperature, wind speed and relative humidity. This line is included in section 4 of the manuscript:

Figures 21, 22 and 23 demonstrate the spatial pattern of temperature, precipitation, relative humidity and wind speed respectively during 1990-2019 from the G2DC-PL+ dataset.

Additionally, the manuscript should include some discussion of the useful spatial scales of the gridded product (2 km scale for all parameters) in view of the much lower station density for most parameters (except precipitation), see specific comments.

**Response**: We agree with the referee that the final 2 km resolution of the product does not match well with the low station density for all variables except precipitation. However, in our view an increase in the spatial resolution of the precipitation dataset from 5 to 2 km is an important and useful thing for many applications, particularly in hydrology. A number of gridded precipitation datasets at comparably high resolution were issued recently (Duan et al., 2016; Lewis et al., 2018; Laiti et al., 2018).

The main reason for setting a 2 km resolution for other variables despite a lower station density is to maintain the same resolution throughout the whole dataset, which should in our view render it more practical for the potential users, especially those who would use our product as forcing data in their environmental models.

Another reason was that temperature, humidity and wind speed were interpolated using kriging with external drift method, in which elevation was used as a co-variable. Elevation at 2 km is much more accurate than at 5 km resolution, especially in high altitude areas, so this should be a clear, although indirect, benefit of this approach.

These considerations were added to the subsection "Spatial resolution". See also the last response to this review.

Specific comments As far as I can see, some improvement of the English grammar is needed in a few places.

1) Space is missing between 2 words at line number 10, 77, 88, 186

Response: Done.

2) Line 23 [refs] needs to be specified.

Response: Done.

3) Line 33 applicatons -> applications

Response: Done.

4) Line 47 remove "ofrange"

Response: Done.

5) Line 65 remove comma+space in "interpolation, ."

Response: Done.

6) Line 77 "will help better constrain hydrological models" rephrase?

**Response:** Rewritten as help to constrain hydrological models

7) Line 82 "kirigging" correct

Response: Done.

8) Line 91 "evap-otransipration" correct.

Response: Done.

9) Line 120 "under-catch" -> "undercatch" for consistent naming in the manuscript..

Response: Done.

10) Line 124 "(Berezowski et al., 2016)" -> "Berezowski et al. (2016)"?

Response: Done.

11) Line 139-140 I guess that "[-]" means without unit, should in my opinion be modified or removed to avoid confusion for the reader.

**Response**: We agree it was confusing, it was removed from the text.**

2.4 Number of stations. This section or the concluding section should in my opinion be supplemented with some discussion of the useful spatial accuracy of the final 2 km data in relation to the station density for the different parameters.

**Response**: Thank you for this comment. We have now realized that subsection order in section 2 was confusing. We have now placed section "Spatial resolution" after the section "Number of stations", since it is more logical to state that the resolution increased as a result of station density increase in comparison to the previous dataset.

As per discussion of the spatial resolution of the dataset vs. station density, see our response to the second comment of this review above.

**References**

Belo-Pereira, M., Dutra, E., and Viterbo, P. (2011), Evaluation of global precipitation data sets over the Iberian Peninsula, J. Geophys. Res., 116, D20101, doi:10.1029/2010JD015481.

Berndt, C., Haberlandt, U. Spatial interpolation of climate variables in Northern Germany—Influence of temporal resolution and network density. Journal of Hydrology: Regional Studies, 15:184-202, 2018

Dobrowski, S. Z., Abatzoglou, J., Greenberg, J., and Schladow, S.: How much influence does landscapescale physiography have on air temperature in a mountain environment?, Agricultural and Forest Meteorology, 149, 1751–1758, https://doi.org/10.1016/j.agrformet.2009.06.006, 2009.

Duan, Z., Liu, J., Tuo, Y., Chiogna G., Disse, M. Evaluation of eight high spatial resolution gridded precipitation products in Adige Basin (Italy) at multiple temporal and spatial scales. Science of The Total Environment, 573, 1536-1553, https://doi.org/10.1016/j.scitotenv.2016.08.213

Hayhoe, K., Wake, C., Anderson, B., Zhong-Liang, X., Maurer, E., Zhu, J., Bradbury, J., DeGaetano, A., Stoner, A., and Wuebbles, D.: Regional climate change projections for the Northeast USA, Mitigation and Adaptive Strategies for Global Change, 13, 425–436, https://doi.org/10.1007/s11027-007-9133-2, 2008.

Laiti, L., Mallucii, S., Piccolroaz, S., Bellin, A., Zardi, D., Fiori, A., Nikulin, G., Majone, B. Testing the Hydrological Coherence of High-Resolution Gridded Precipitation and Temperature Data Sets. Water Resources Research, 54, 3, 1999-2016, https://doi.org/10.1002/2017WR021633

Lewis, E., Quinn, E., Blenkinsop, S., Fowler, H., Freer, J., Tanguy, M., Hitt, O., Coxon, G., Bates, P., Woods, R. A rule based quality control method for hourly rainfall data and a 1 km resolution gridded hourly rainfall dataset for Great Britain: CEH-GEAR1hr. Journal of Hydrology. 564, 930-943, https://doi.org/10.1016/j.jhydrol.2018.07.034

---

## Referee Report (RR1)

The manuscript "G2DC-PL+ A gridded 2 km daily climate dataset for the union of the Polish territory and the Vistula and Odra basins" by Piniewski et al , submitted to Earth System Science Data presents a second, updated version of a high-resolution gridded daily precipitation, temperature, relative humidity and wind speed  dataset as well as its description.

Overall, I assess very positively the main achievement of the paper – freely available dataset of high resolution daily data of containing key factors for calculating many climate parameters.

The manuscript is well organized and written quite clear. The reference list is complete. The methodology of interpolation method is based on the work by Berezowski et. al and only the differences between both approaches are described here. The description of the evaluation of interpolation errors was significantly improved.

I am satisfied with the corrections made by the authors and in my opinion the paper can be published in this version (3). However I still feel some the deficiency in the discussion of the effect of station density on interpolation errors.